# An Arginine-Rich Motif in the ORF2 capsid protein regulates the hepatitis E virus lifecycle and interactions with the host cell

**Kévin Hervouet**[1☯], **Martin Ferrié**[1☯], **Maliki Ankavay**[1,2☯], **Claire Montpellier**[1], **Charline Camuzet**[1], **Virginie Alexandre**[1], **Aïcha Dembélé**[1], **Cécile Lecoeur**[1], **Arnold Thomas Foe**[1], **Peggy Bouquet**[3], **David Hot**[3], **Thibaut Vausselin**[1], **Jean-Michel Saliou**[3], **Sophie Salomé-Desnoulez**[3], **Alexandre Vandeputte**[3], **Laurent Marsollier**[4], **Priscille Brodin**[1,3], **Marlène Dreux**[5], **Yves Rouillé**[1], **Jean Dubuisson**[1], **Cécile-Marie Aliouat-Denis**[1], **Laurence Cocquerel**[1] *

1 University of Lille, CNRS, INSERM, CHU Lille, Pasteur Institute of Lille, U1019-UMR 9017-CIIL- Center for Infection and Immunity of Lille, Lille, France, 2 Division of Gastroenterology and Hepatology, Institute of Microbiology, Lausanne, Switzerland, 3 Univ. Lille, CNRS, Inserm, CHU Lille, Institut Pasteur de Lille, UMR2014 - US41 - PLBS-Plateformes Lilloises de Biologie & Santé, Lille, France, 4 Université d'Angers, Nantes Université, INSERM, Immunology and New Concepts in ImmunoTherapy, INCIT, UMR 1302, Angers, France, 5 CIRI - Centre International de Recherche en Infectiologie, Univ Lyon, Université Claude Bernard Lyon 1, Inserm-U1111, CNRS-UMR5308, ENS-Lyon, Lyon, France

☯ These authors contributed equally to this work.
* laurence.cocquerel@cnrs.fr

**Data Availability Statement:** All relevant data are within the manuscript and its Supporting information files.

## Abstract

Hepatitis E virus (HEV) infection is the most common cause of acute viral hepatitis worldwide. Hepatitis E is usually asymptomatic and self-limiting but it can become chronic in immunocompromised patients and is associated with increased fulminant hepatic failure and mortality rates in pregnant women. HEV genome encodes three proteins including the ORF2 protein that is the viral capsid protein. Interestingly, HEV produces 3 isoforms of the ORF2 capsid protein which are partitioned in different subcellular compartments and perform distinct functions in the HEV lifecycle. Notably, the infectious ORF2 (ORF2i) protein is the structural component of virions, whereas the genome-free secreted and glycosylated ORF2 proteins likely act as a humoral immune decoy. Here, by using a series of ORF2 capsid protein mutants expressed in the infectious genotype 3 p6 HEV strain as well as chimeras between ORF2 and the CD4 glycoprotein, we demonstrated how an Arginine-Rich Motif (ARM) located in the ORF2 N-terminal region controls the fate and functions of ORF2 isoforms. We showed that the ARM controls ORF2 nuclear translocation likely to promote regulation of host antiviral responses. This motif also regulates the dual topology and functionality of ORF2 signal peptide, leading to the production of either cytosolic infectious ORF2i or reticular non-infectious glycosylated ORF2 forms. It serves as maturation site of glycosylated ORF2 by furin, and promotes ORF2-host cell membrane interactions. The identification of ORF2 ARM as a unique central regulator of the HEV lifecycle uncovers how viruses settle strategies to condense their genetic information and hijack cellular processes.

**Funding:** This work was supported by grants from the French agency ANRS-Maladies infectieuses émergentes (ECTZ61515, AAP2015-2 CSS4), Pasteur Institute of Lille (#Diag-HepE), Région Hauts-de-France (#Diag-HepE) and Inserm-transfert (#activVHE). The PLBS platform used in this work was supported by the ANR (ANR-10-EQPX-04-01) and Feder (12001407 [D-AL] EquipEx ImagInEx BioMed). K.H., M.A. and T.V. were supported by a fellowship from the ANRS-Maladies infectieuses émergentes (ECTZ61515, AAP2015-2 CSS4). M.F. was supported by a fellowship from the Pasteur Institute of Lille and Région Hauts-de-France. M.A. is currently supported by the Fonds National Suisse (FNS). The funders had no role in study design, data collection and analysis, decision to publish, or preparation of the manuscript.

**Competing interests:** The authors have declared that no competing interests exist.

## Author summary

Hepatitis E virus (HEV) is the major cause of acute viral hepatitis worldwide. Although infection with HEV is usually self-resolving, it can cause up to 30% mortality in pregnant women in the third trimester. There is no specific treatment nor universal vaccine to fight against HEV. In our study, we focused on the HEV ORF2 capsid protein which is produced as different forms that perform distinct functions in the HEV lifecycle. The infectious ORF2i form is the structural component of virions, while the other forms likely act as an immune decoy. Herein, we deciphered the molecular determinants of ORF2 multifunctionality. We identified an Arginine-Rich Motif (ARM) located in the ORF2 N-terminal region that controls the subcellular localization, the fate and functions of ORF2 forms. It also promotes ORF2-host cell interactions. Our observations highlight the ORF2 ARM as a unique central regulator of ORF2 addressing that finely controls the HEV lifecycle.

## Introduction

Hepatitis E virus (HEV) is the most common cause of acute viral hepatitis worldwide and is an emerging problem in industrialized countries. This virus causes about 20 million infections annually [1]. While HEV infection is asymptomatic for most patients, some human populations including pregnant women and immunocompromised patients have higher risk to develop severe forms and chronic infections, respectively. HEV strains infecting humans have been classified into 4 main distinct genotypes (gt) belonging to a single serotype [2]. Gt1 and gt2 that infect humans only, are primarily transmitted through contaminated drinking water and are responsible for waterborne hepatitis outbreaks in developing countries. In contrast, gt3 and gt4 are zoonotic and are largely circulating in industrialized countries. They are mainly transmitted by contact with swine and consumption of inadequately heated pork products [3]. There is no specific treatment nor universal vaccine to fight against HEV.

HEV is a quasi-enveloped [4,5], positive-sense RNA virus expressing three open reading frames (ORFs): ORF1, ORF2 and ORF3 [6]. ORF1 encodes the ORF1 non-structural polyprotein that is the viral replicase [7]. ORF2 encodes the ORF2 viral capsid protein and ORF3 encodes a small protein that is involved in virion morphogenesis and egress [8,9]. Since studying the HEV lifecycle has long been hampered by the absence of efficient systems to amplify HEV, many steps of the HEV lifecycle remain poorly understood [10]. By combining the gt3 p6 strain [11] and a highly transfectable subclone of PLC/PRF/5 cells (PLC3 cells), we previously described an efficient HEV cell culture system [12]. This model notably enabled the pioneering demonstration that, during its lifecycle, HEV produces at least 3 forms of the ORF2 capsid protein: infectious ORF2 (ORF2i), glycosylated ORF2 (ORF2g), and cleaved ORF2 (ORF2c). The ORF2i protein is the structural component of infectious particles. It is not glycosylated and is likely derived from the assembly of the intracellular ORF2 (ORF2intra) form present in the cytosolic compartment. Importantly, we showed that a fraction of the ORF2intra form is translocated into the nucleus of infected cells [13]. Nuclear localization of ORF2 was also described by Lenggenhager *et al.* in naturally infected human hepatocytes [14], but the significance of ORF2 nuclear localization remains to be elucidated. The ORF2g and ORF2c proteins (herein referred to as ORF2g/c) are highly glycosylated and secreted in large amounts in culture supernatant (*i.e.*, about 1000x more than ORF2i [15]) and are the most abundant antigens detected in patient sera [12]. These proteins likely act as a humoral immune decoy that inhibits antibody-mediated neutralization [15]. How the different forms of ORF2 are

generated during the HEV lifecycle has not yet been fully investigated. However, their sequence and post-translational modifications suggest that they might be produced either by a distinct addressing into the secretory pathway and the nucleus [12,13], and/or by a differential translation process [15].

Here we investigated the mechanisms by which the ORF2 forms are produced and differentially addressed to cell compartments. We demonstrated that HEV has set up a nucleo-cytoplasmic transport mechanism of its capsid protein to modulate cell host immune responses. In addition, we found that during the HEV lifecycle, a fine-tuning of ORF2 partitioning occurs between cytosolic, reticular and nuclear compartments. Importantly, we identified a stretch of 5 amino acid residues (herein referred to as ARM, Arginine-Rich Motif) in the N-terminal region of the ORF2 protein that drives nuclear translocation and tightly modulates the stoichiometry between the different ORF2 forms, especially by regulating the functionality of the ORF2 signal peptide and interactions with the host cell.

## Results

### The ORF2 protein transits through the nucleus in the early phase of replication and infection

We and others previously showed that the ORF2 protein is translocated into the nucleus of infected cells of patient liver biopsies [14] and in cell culture system [13]. Here, we first analyzed by immunofluorescence the ORF2 expression in PLC3 cells electroporated with the infectious gt3 p6 strain (PLC3/HEV-p6), and the infectious gt1 Sar55 strain (PLC3/HEV-Sar55) at different time points post-electroporation (p.e.) (Fig 1A and 1B). ORF2 staining and quantification of nuclear/cytosolic fluorescence ratio showed that ORF2 displays a nuclear localization at early time points p.e. (*i.e.*, 18h) that decreases over time. A similar subcellular localization was observed in PLC3 cells electroporated with another gt3 strain (HEV83-2, Fig 1C) and Huh-7.5 cells infected with HEV-p6 particles (Fig 1D). Note that in these cells, the ORF2 expression was delayed but displayed a similar profile. In addition, at 6 days (6d) post-infection (p.i.) Huh-7.5 cells displayed different ORF2 populations likely corresponding to newly infected cells (Fig 1D). Taken together, these results indicate that nuclear translocation of ORF2 takes place at early time points and is then followed by a nuclear export process, indicating that HEV has developed mechanisms for ORF2 nuclear import and export.

### The ORF2 protein displays an Arginine-Rich Motif (ARM) that functions as a Nuclear Localization Signal (NLS)

To decipher the molecular mechanisms of ORF2 nuclear import, we first analyzed its amino acid (aa) sequence with the NLSTradamus prediction program [16]. We identified a potential Nuclear Localization Signal (NLS) corresponding to a conserved Arginine-Rich Motif (ARM, 5 arginine residues: RRRGRR) in the N-terminal region of ORF2, downstream of its signal peptide (SP) (Fig 2A and S1 Fig). We next generated a series of ORF2 mutants in the p6 strain that are depicted in Fig 2A. We characterized their expression and subcellular localization (Fig 2B and 2C, S2 and S3 Figs), and their impact on the HEV lifecycle (Fig 2D) in PLC3/HEV-p6 cells.

The replacement of arginine by alanine residues (3R/3A, 2R/2A and 5R/5A mutants) led to a drastic reduction of ORF2 nuclear localization compared to the wt protein (Fig 2B and 2C Nuclear extract, and S2 Fig), indicating that the ARM is likely a functional NLS. Interestingly, the reduced nuclear localization of these mutants was associated with an accumulation of ORF2 in the Golgi apparatus (S3 Fig) and a reduced association with cellular membranes (Fig

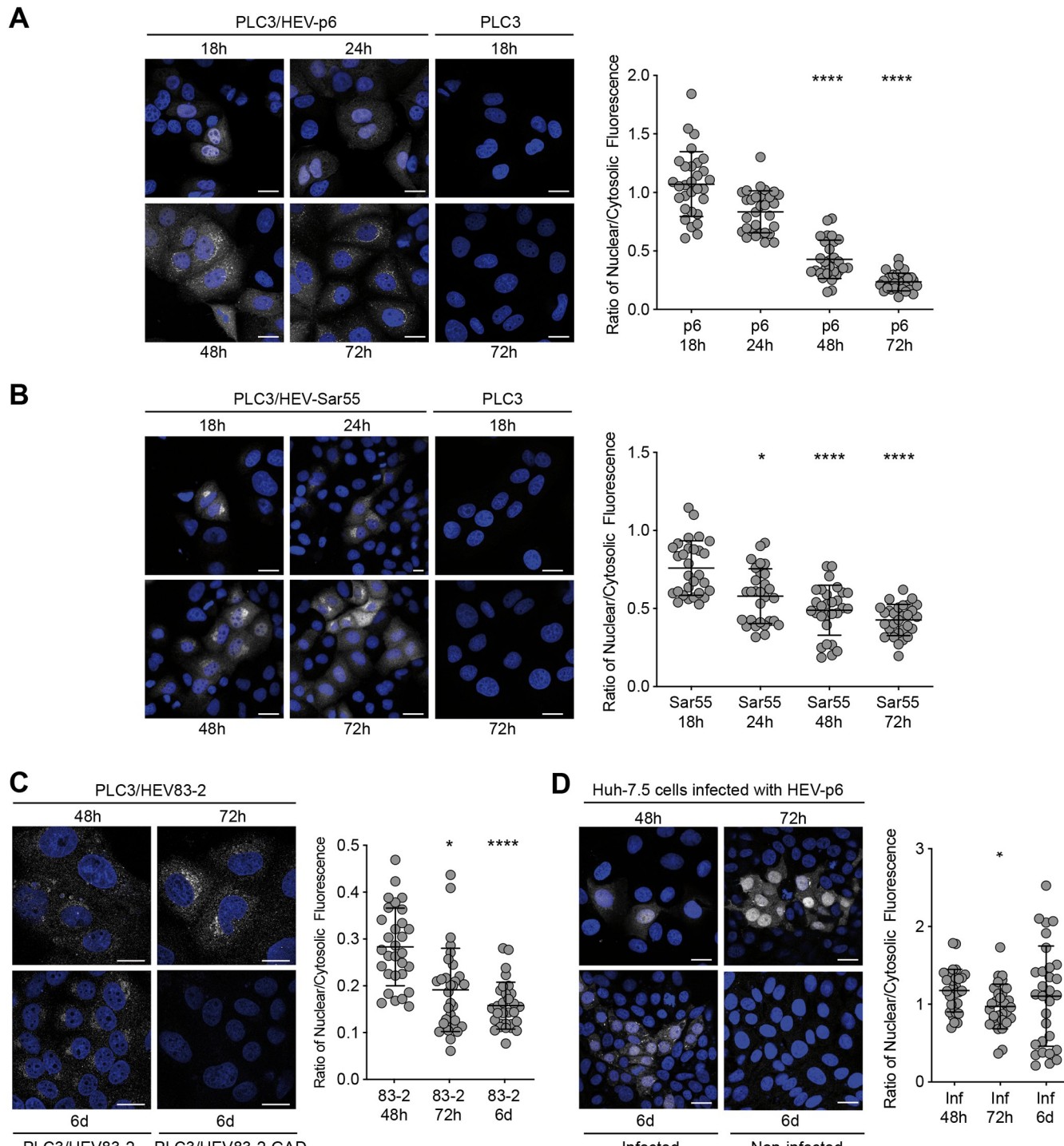

**Fig 1. Kinetics of the subcellular localization of ORF2 protein.** PLC3 cells electroporated with HEV-p6 (A), HEV-Sar55 (B) and HEV83-2 (C) RNA and Huh-7.5 cells infected with HEV-p6 (D) were fixed at the indicated timepoints post-electroporation (p.e.) (A-C) or post-infection (p.i.) (D). Indirect immunofluorescence analysis was performed using the 1E6 anti-ORF2 antibody. Cells were analyzed by confocal microscopy (magnification x40). Scale bar, 20 μm. Nuclear/cytosolic fluorescence intensity quantification was done using ImageJ software (mean ± S.D., $n \geq 30$ cells, Friedman with Nemenyi test). *p < 0.05, ****$p < 0.0001$. Mock electroporated PLC3 cells (PLC3), cells electroporated with a replication-deficient HEV83-2 strain (PLC3/HEV83-2-GAD) and non-infected Huh-7.5 cells were used as controls.

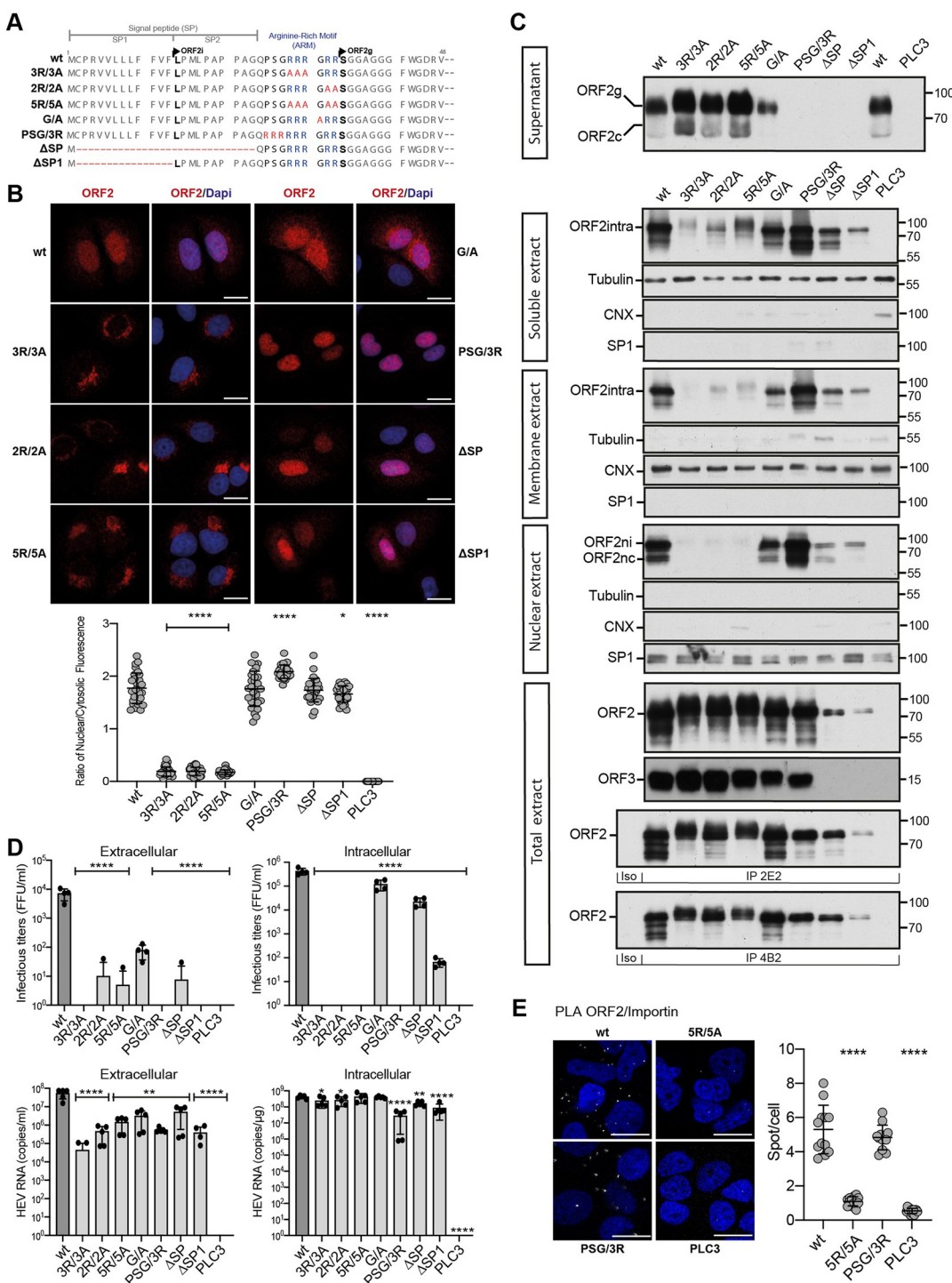

**Fig 2. ORF2 contains an Arginine-Rich Motif (ARM) that is important for its nuclear localization.** (A) Schematic sequence alignment of HEV-p6 ORF2wt and ARM/SP mutants. (B) Subcellular localization of HEV-p6 ORF2wt and ARM/SP mutants. PLC3 cells were electroporated with wt and mutant HEV-p6 RNAs. At 18 h.p.e, cells were processed for indirect immunofluorescence using the 1E6 anti-ORF2 antibody (Ab) and analyzed by confocal microscopy (magnification x63). Red = ORF2; Blue = DAPI. Scale bar, 20μm. Nuclear/cytosolic fluorescence intensity quantification was done using ImageJ software (mean ± S.D., $n \geq 30$ cells, Kruskal-Wallis with Conover's test). $^*p < 0.05$, $^{****}p < 0.0001$. (C) Subcellular fractionation of PLC3/HEV-p6 expressing ORF2wt and ARM/SP mutants at 10 d.p.e. Fractionation was done using a subcellular protein fractionation kit for cultured cells. ORF2 proteins were detected by WB with 1E6 Ab. Glycosylated ORF2 (ORF2g), cleaved ORF2 (ORF2c), intracellular ORF2 (ORF2intra), nuclear ORF2intra (ORF2ni), nuclear and cleaved

ORF2intra (ORF2nc) are indicated. ORF3 protein in cell lysates was detected with a rabbit anti-ORF3 Ab. Tubulin, ER marker Calnexin (CNX) and the transcription factor SP1 used as a nuclear marker, were also detected to check the quality of fractionation. 2E2 and 4B2 are conformation-specific anti-ORF2 antibodies. Molecular mass markers are indicated on the right (kDa). (D) Infectious titer determination and HEV RNA quantification in PLC3/HEV-p6 expressing ORF2wt or mutant proteins. Extra- and intracellular viral particles were extracted at 10 d.p.e and used to infect naïve Huh-7.5 cells for 3 days. Cells were next processed for indirect immunofluorescence. ORF2-positive cells were counted and each positive cell focus was considered as one FFU. Results were expressed in FFU/ml ($n = 4$). Extra- and intracellular viral RNAs were quantified at 10 d. p.e by RT-qPCR ($n \geq 5$) (mean ± S.D., Kruskal-Wallis with Conover's test). $^*p < 0.05$, $^{**}p < 0.01$, $^{****}p < 0.0001$. (E) PLC3/ HEV-p6-wt, PLC3/HEV-p6-5R/5A, PLC3/HEV-p6-PSG/3R and PLC3 mock cells were processed for proximity ligation assay using antibodies to ORF2 and Importin-α1 at 18 h.p.e. Stacks of images corresponding to the total volume of the cells were acquired, and maximum intensity projections of the stacks were generated. For each condition, 12 fields of cells were analyzed (total cell number $\geq$ 165). Scaled regions of interest of a representative field (left) and quantification of spot/cell (right) are shown (mean ± S.D., Kruskal-Wallis with Conover's test). $^{****}p < 0.0001$.

2C, Membrane extract), indicating that these mutated proteins are likely soluble in the Golgi lumen. Mutations did not affect the stability or folding of ORF2, as demonstrated by blotting total extracts and immunoprecipitations with conformation-specific anti-ORF2 antibodies (Fig 2C, total extract, IP 2E2 and IP 4B2). Some ORF2 forms with an apparent higher molecular weight were observed in the soluble fraction of 3R/3A and 5R/5A mutants, and an increase of ORF2g/c secretion were observed for the three mutants (Fig 2C), suggesting that ARM mutations induced a higher translocation into the secretory pathway likely associated to an improved functionality of ORF2 SP. Quantification of intracellular RNAs showed that replication was not altered in these mutants (Fig 2D and S4 Fig). In addition, ARM mutations did not affect ORF3 expression (Fig 2C). However, these mutants no longer produced infectious viral progeny (Fig 2D). Thus, our results suggest that the ARM drives the ORF2 nuclear translocation and plays important functions in the HEV lifecycle, notably in the assembly of infectious particles.

We also generated mutants for which Pro25, Ser26 and Gly27 residues were replaced by 3 arginine residues (PSG/3R mutant), and alternatively SP was fully or partially deleted (ΔSP and ΔSP1 mutants, respectively) (Fig 2A). The PSG/3R mutant showed an increased nuclear localization (Fig 2B and 2C Nuclear extract and S2 Fig) but was impaired in ORF2g/c secretion (Fig 2C, Supernatant), as observed for the SP deletion mutants, indicating that the addition of arginine residues strengthens the NLS function of ARM but inhibits the functionality of ORF2 SP. The PSG/3R mutant expressed the ORF3 protein but displayed lower intracellular replication levels and was no longer infectious (Fig 2D). The increased nuclear localization of this mutant is therefore likely responsible for the reduction of HEV RNA replication and assembly of infectious particles.

The full (ΔSP) or partial (ΔSP1) deletion of the ORF2 SP led to a total inhibition of ORF2 secretion (Fig 2C, Supernatant), as expected due to the absence of reticular translocation. Although ΔSP1 ORF2 expression or stability was reduced (Fig 2C), both SP deletion mutants still exhibited a nuclear localization (Fig 2B and S2 Fig), indicating that the nuclear translocation process is independent of the reticular translocation. Because ORF2 and ORF3 are overlapping in the HEV genome, and ORF3 is essential to particle secretion, the SP deletion mutants did not express the ORF3 protein (Fig 2C) and displayed reduced extracellular titers (Fig 2D). Intracellular titers were also lowered in SP mutants (Fig 2D), indicating that the ORF2 SP likely plays an important role in replication or assembly of infectious particles.

Lastly, the highly conserved Gly31 residue (S1 Fig) was also mutated (Fig 2A, G/A mutant). This mutant displayed a subcellular distribution similar to that of wt, and expressed the ORF3 protein. Although to a lesser extent than wt, G/A mutant produced intracellular particles, but showed reduced extracellular RNA and infectious levels. This indicates that the G/A mutation affects particle secretion.

We next carried out a comparative study of NLS sequences in viral proteins and their importin. We found that the ORF2 ARM is similar to the Epstein-Barr nuclear antigen leader protein (EBNA-LP) arginine-rich NLS (RRVRRR) that interacts with Importin-α1 [17]. Interestingly, ORF2 and Importin-α1 co-localized in the nucleus of infected cells with a Pearson correlation coefficient (PCC) of 0.670, and the mutation of arginine residues drastically reduced this colocalization (S5 Fig). In addition, we performed a proximity ligation assay (PLA) using ORF2 and Importin-α1 antibodies in PLC3/HEV-p6-wt, PLC3/HEV-p6-5R/5A, PLC3/HEV-p6-PSG/3R and PLC3 mock cells (Fig 2E). In cells expressing ORF2 wt or the PSG/3R mutant, many dots were observed whereas in mock cells or cells expressing the 5R/5A mutant, very few fluorescent dots were observed. Taken together, our results indicate that ORF2 likely interacts with Importin-α1 thanks to its ARM that serves as a functional NLS. In addition, these results suggest that the ARM is involved in the fine-tuning of the addressing and stoichiometry of the ORF2 protein between the nuclear, cytosolic and reticular pathways. This stoichiometry is likely essential to the HEV lifecycle.

## ORF2 nuclear translocation modulates host gene expression

Our results demonstrated that ORF2 localizes in the nucleus (ORF2ni) and the ARM is pivotal for nuclear translocation. Our results also suggested that ORF2ni is readily detected as early as 18 h.p.e. in PLC3/HEV-p6 and PLC3/HEV-Sar55 cells while its nuclear targeting is transient and starts to decrease after 48h (Fig 1A and 1B). This observation prompted us to address the impact of early nuclear translocation of ORF2 on the regulation of host genes. We performed a transcriptomic analysis by microarrays (Agilent SurePrint Technology) in PLC3/HEV-p6-wt, PLC3/HEV-p6-5R/5A, PLC3/HEV-p6-ΔORF3 and PLC3 mock cells at 18 h.p.e. (S6 Fig). PLC3/HEV-p6-ΔORF3 express an ORF3-null mutant of HEV-p6 (S6C and S7A Figs). Interestingly, in PLC3/HEV-p6-wt and PLC3/HEV-p6-ΔORF3 cells, we observed a significant inhibition of expression of 7 genes related to the TNFα, IL-17 and NF-κB-mediated signaling as well as inflammatory responses (i.e., NOD-like receptor-induced response) (S6 Fig). In contrast, no gene expression inhibition was observed in PLC3/HEV-p6-5R/5A cells, reflecting the importance of ORF2 nuclear translocation in the observed inhibition. In addition, while some reports suggested that ORF3 expression modulates the host responses [18–23], no marked difference was observed when comparing ΔORF3 mutant to wt. Of note, kinetics of ORF2 nuclear localization in cells expressing the ΔORF3 mutant was similar to wt (S7B Fig), indicating that the nucleo-cytoplasmic transport of ORF2 is an ORF3-independent process.

Altogether our results suggest that the ORF2 ARM, which notably regulates ORF2 nuclear translocation, is a pivotal viral determinant for the modulation of host pathways and, especially, genes of the NF-κB-induced signaling upon infection. Further studies will be required to define precisely the impact of this HEV-driven host regulation on immune cell responses.

## Nuclear export of the ORF2 protein

The observation that ORF2 nuclear targeting is transient and decreases with time (Fig 1) then prompted us to investigate the mechanisms of ORF2 nuclear export. We treated PLC3/HEV-p6 (Fig 3A), PLC3/HEV83-2 and HEV-p6 infected Huh-7.5 (S8A Fig) cells with nuclear export inhibitors, Leptomycin B (LepB) and Verdinexor (Verd). These compounds are irreversible (LepB) and reversible (Verd) inhibitors of the ubiquitous transport receptor chromosome maintenance protein 1 (CRM1/Exportin 1), which recognizes hydrophobic leucine-rich export signals [24]. Treated cells displayed a highly significant nuclear accumulation of ORF2, as compared to control cells (Figs 3A and S8A). Co-localization studies revealed that ORF2 partially or transiently colocalizes with CRM1 in untreated cells whereas they significantly

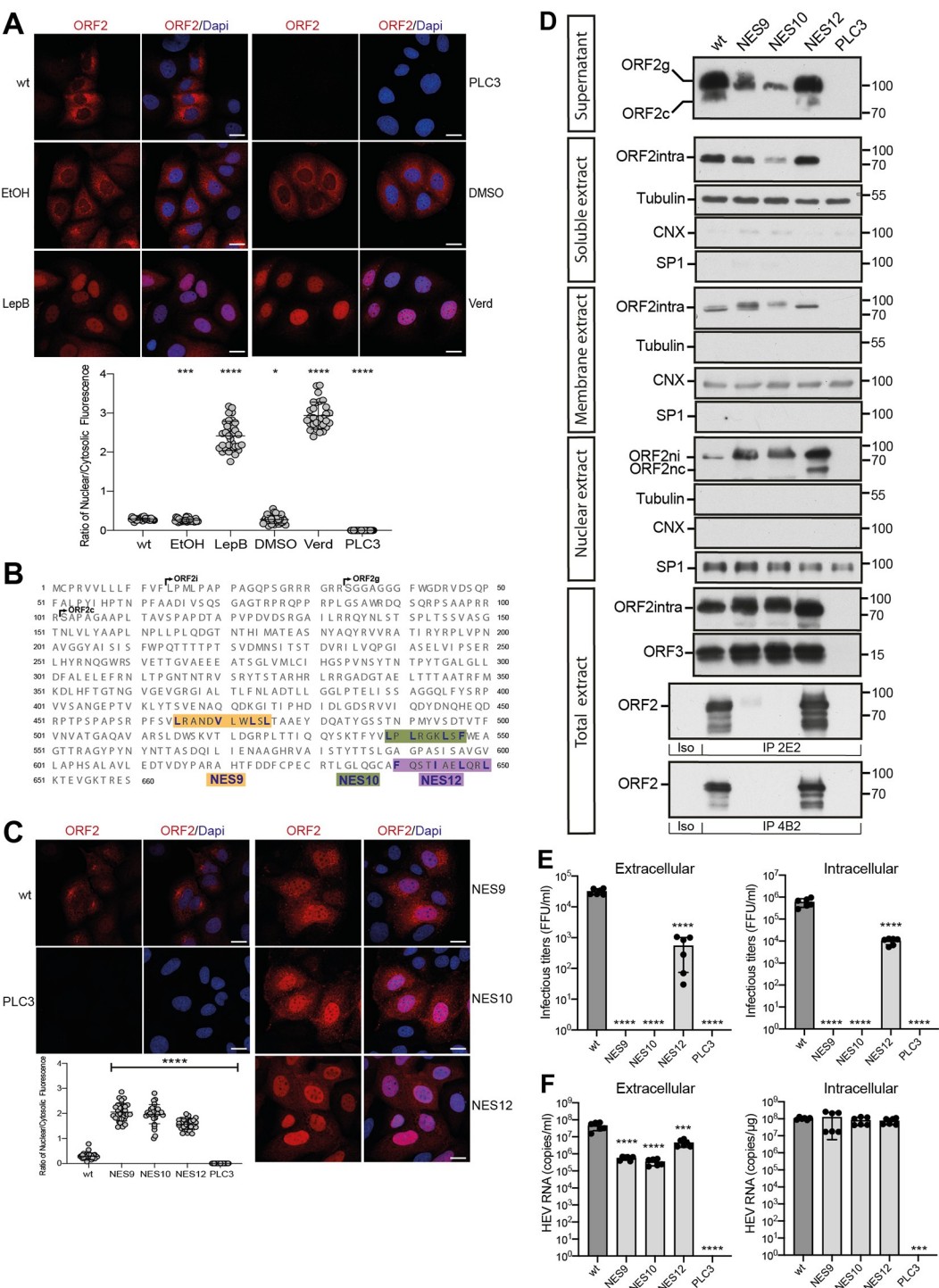

**Fig 3. Nuclear export of the ORF2 protein.** (A) Analysis of ORF2 nuclear export in inhibitors treated-PLC3/HEV-p6 cells. Cells were treated at 32 h.p.e with 20nM of Leptomycin B (LepB), 100nM of Verdinexor (Verd) or diluent (EtOH or DMSO, respectively) for 16h. Cells were processed for indirect immunofluorescence with the 1E6 anti-ORF2 Ab and analyzed by confocal microscopy (magnification x63). Red = ORF2; Blue = DAPI. (B) Schematic representation of HEV-p6 ORF2 protein sequence highlighting the three studied NES motifs (i.e., NES9, NES10 and NES12). (C) Subcellular localization of ORF2 NES mutants at 48 h.p.e. Red = ORF2; Blue = DAPI. In A and C, the scale bars correspond to 20μm, and nuclear/cytosolic fluorescence intensity quantification was done using ImageJ software (mean ± S.D., $n \geq 30$ cells, Kruskal-Wallis with Conover's test). $^{*}p < 0.05$, $^{***}p < 0.001$, $^{****}p < 0.0001$. (D) Subcellular fractionation of PLC3/HEV-p6 expressing ORF2wt and NES mutants at 4 d.p.e. Fractionation was done using a subcellular protein fractionation kit for cultured cells. ORF2

proteins were detected by WB with 1E6 Ab. Glycosylated ORF2 (ORF2g), cleaved ORF2 (ORF2c), intracellular ORF2 (ORF2intra), nuclear ORF2intra (ORF2ni), nuclear and cleaved ORF2intra (ORF2nc) are indicated. ORF2 and ORF3 proteins in total cell lysates were detected with 1E6 Ab and a rabbit anti-ORF3 Ab, respectively. Tubulin, ER marker Calnexin (CNX) and the transcription factor SP1 used as a nuclear marker, were also detected to check the quality of fractionation. 2E2 and 4B2 are conformation-specific anti-ORF2 antibodies. Molecular mass markers are indicated on the right (kDa). (E) Infectious titer determination in PLC3/HEV-p6 expressing ORF2wt or NES mutants. Extra- and intracellular viral particles were extracted at 10 d.p.e and used to infect naïve Huh7.5 cells for 3 days. Cells were next processed for indirect immunofluorescence. ORF2-positive cells were counted and each positive cell focus was considered as one FFU. Results were expressed in FFU/ml. (F) HEV RNA quantification in PLC3/HEV-p6 expressing ORF2wt or NES mutants. Extra- and intracellular viral RNAs were quantified at 10 d.p.e by RT-qPCR. In E and F, $n = 6$, mean ± S.D., Kruskal-Wallis with Conover's test. ***$p < 0.001$, ****$p < 0.0001$.

colocalize upon treatment with nuclear export inhibitors (S8B Fig). In addition, PLA experiments with anti-ORF2 and anti-CRM1 showed that ORF2 likely interacts with CRM1 (S8C Fig). These results indicate that ORF2 undergoes a nuclear export to the cytoplasm by a CRM1-dependent mechanism.

CRM1 recognizes hydrophobic leucine-rich export signals. By analyzing the ORF2 sequence, we identified at least 12 potential nuclear export signals (NES) into the ORF2 sequence. We replaced the hydrophobic residues in these motifs by alanine residues and characterized the generated mutants as described above. Most mutations did not affect ORF2 partitioning, whereas mutations of three conserved motifs (S9–S11 Figs), named NES9, NES10 and NES12 (Fig 3B) led to a highly significant accumulation of ORF2 inside the nucleus (Fig 3C and 3D) and colocalization with CRM1 (S8B Fig), as observed for cells treated with nuclear export inhibitors (Fig 3A and S8B Fig). Although intracellular replication was not altered by NES mutations (Fig 3F), NES9 and NES10 mutants were no longer infectious, and the NES12 mutant exhibited highly reduced intracellular and extracellular titers (Fig 3E). By immunoprecipitation with conformation-specific anti-ORF2 antibodies, we found that NES9 and NES10 mutants were misfolded whereas NES12 mutant was properly folded (Fig 3D, IP 2E2 and IP 4B2). These results indicate that NES9 and NES10 motifs are functional NES but their mutation affect ORF2 protein folding and thus particle assembly. In contrast, the reduced infectious particle assembly of NES12 mutant is likely due solely to its differential subcellular localization (S7C Fig), and not to a defect in viral replication or ORF2 folding. Of note, the NES12 mutant showed a reduced interaction with CRM1 (S8C Fig).

Thus, HEV has set up a nuclear export system for its ORF2 capsid protein. This mechanism involves CRM1 which likely recognizes three conserved NES on the ORF2 sequence. Moreover, these results suggest again that a fine balance between the nuclear, cytosolic and reticular pathways is likely essential to the HEV lifecycle.

## Translocation and maturation of the glycosylated ORF2 forms

Next, we investigated the mechanisms of translocation and maturation of the highly secreted and glycosylated ORF2g/c isoforms. First, we treated PLC3/HEV-p6 and Mock cells with Mycolactone, an inhibitor of Sec61 translocon, the membrane embedded protein complex responsible for the translocation of newly synthetized polypeptides into the ER lumen [25]. Interestingly, we observed a dose-dependent reduction of ORF2g/c secretion in Mycolactone-treated PLC3/HEV-p6 cell supernatants (Fig 4A), indicating that reticular translocation of the ORF2g/c forms is Sec61-dependent.

Previously, we demonstrated that the first residues of ORF2i, ORF2g and ORF2c proteins are Leu14, Ser34 and Ser102, respectively [12,13] (Fig 3B). Therefore, the first 20 aa of the ORF2i protein are not present in the ORF2g/c isoforms. Furthermore, the ORF2i protein is

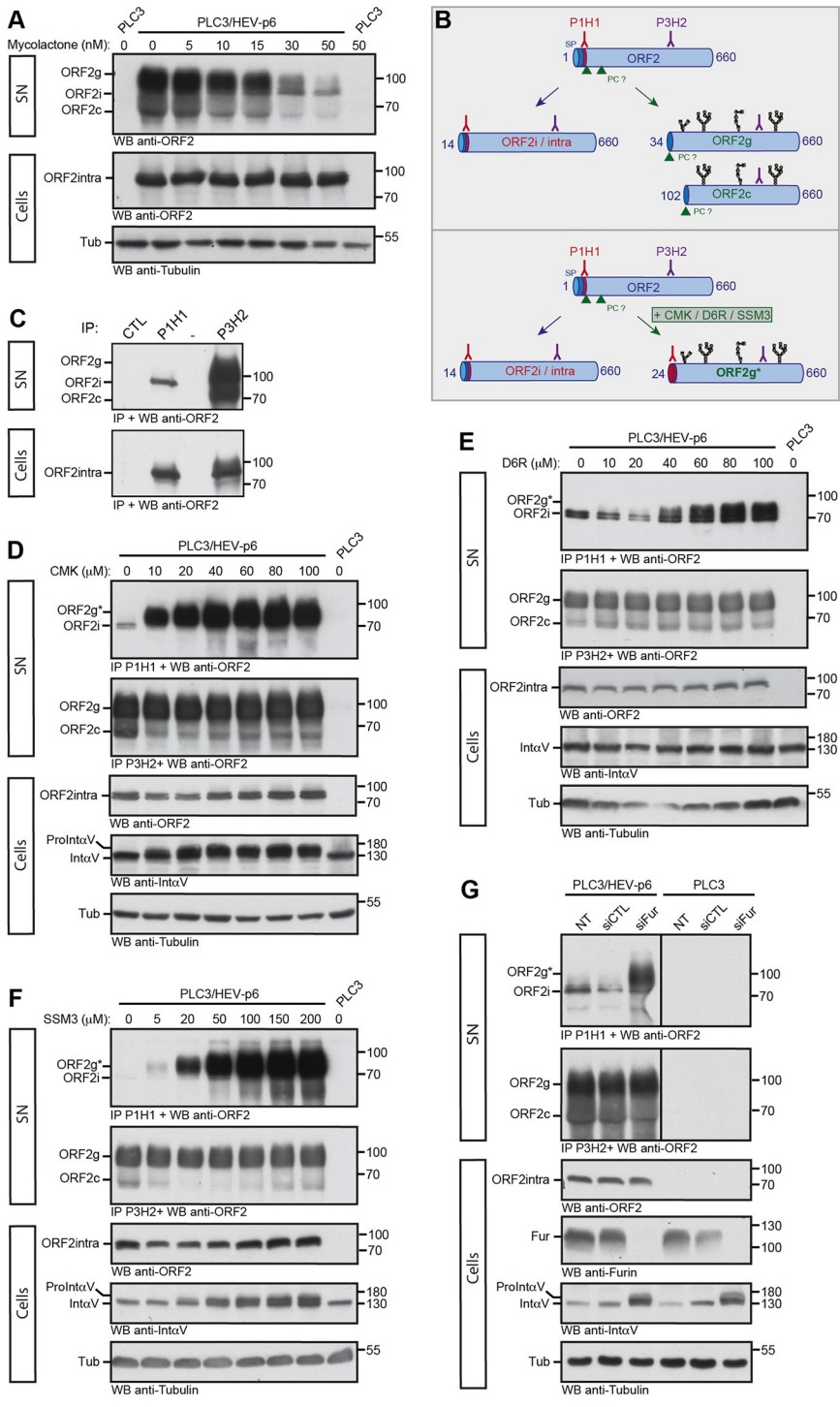

**Fig 4. Translocation and maturation of the glycosylated ORF2 forms.** (A) Dose-response inhibition of ORF2g/c secretion in mycolactone-treated cells. At 7 d.p.e., PLC3/HEV-p6 and mock cells were treated for 24h with the indicated concentrations of mycolactone (in nM) or maximal volume of the vehicle, ethanol (indicated as 0 nM). Supernatants (SN) and lysates (Cells) were collected and ORF2 proteins were detected by WB using the 1E6 Ab. Tubulin served as control protein loading. (B) Schematic representation of ORF2i/g/c proteins and recognition sites of P1H1 and P3H2 antibodies used to discriminate the different ORF2 forms. SP, signal peptide. PC, proprotein convertase. Glycans are in black. (C) Immunoprecipitation of ORF2 proteins in SN and lysates of PLC3/HEV-p6 cells by P1H1, P3H2 and isotype control (CTL) antibodies immobilized on magnetic beads. ORF2 proteins were detected by WB using the 1E6 Ab. (D-F) At 7 d.p.e., PLC3/HEV-p6 cells were treated with the indicated concentrations of

Decanoyl-RVKR-chloromethylketone (CMK), hexa-D-arginine amide (D6R) and SSM3 trifluoroacetate (SSM3) (in μM) or DMSO diluent (indicated as 0 μM). (G) At 7 d.p.e., PLC3/HEV-p6 and PLC3 mock cells were transfected with siRNA targeting furin (siFur) or non-targeting siRNA (siCTL) or left non-transfected (NT). (D-G) At 72h post-treatment or post-transfection, supernatants (SN) and lysates (Cells) were collected. SN were immunoprecipitated with P1H1 and P3H2 antibodies and ORF2 proteins were detected by WB using the 1E6 Ab. ORF2intra, αV-Integrin (IntαV) and Tubulin (Tub) were detected in cell lysates. In G, furin (Fur) was also detected in cell lysates. αV-pro-integrin (ProintαV) corresponds to the non-matured αV-integrin. ORF2g* corresponds to the ORF2g immunoprecipitated by the P1H1 Ab. Molecular mass markers are indicated on the right (kDa).

not glycosylated whereas ORF2g/c proteins are highly glycosylated [13] (Fig 4B). Thanks to these features, we generated a murine monoclonal antibody (P1H1) that recognizes the N-terminus of ORF2i (Fig 4B). P1H1 specifically immunoprecipitates the ORF2i protein without cross-reacting with the highly secreted and glycosylated ORF2g/c proteins (Fig 4C, SN). We also generated the P3H2 antibody that recognizes the different isoforms of ORF2. Both antibodies recognize the intracellular ORF2 form (Fig 4C, Cells).

These antibodies were used to evaluate the effects of three furin inhibitors and related pro-protein convertases (PC) on ORF2g/c maturation. PC cleave the multibasic motifs $R/K-X_n-R/K$ (where $X_n$ corresponds to a 0-, 2-, 4-, 6-amino-acid spacer) in the precursor proteins [26]. The presence of ARM and RRR motif upstream of the ORF2g/c N-termini (Fig 3B), respectively, suggests that a PC might be involved in the maturation of these ORF2 forms. Therefore, we treated PLC3/HEV-p6 cells with three potent furin/PC inhibitors (decanoyl-RVKR-chloromethylketone [CMK], hexa-D-arginine amide [D6R], and SSM3 trifluoroacetate [SSM3]) [27] and immunoprecipitated ORF2 proteins in cell supernatants with P1H1 and P3H2 antibodies. Intracellular contents were probed by WB for ORF2intra, cleavage of cellular αV-pro-integrin (a substrate of intracellular furin) and tubulin (Fig 4D–4F). In these experiments, immunoprecipitation of ORF2g by P1H1 antibody was used as a read-out of the inhibition of ORF2g maturation (Fig 4B, ORF2g*). In treated cells, we observed a dose-dependent immunoprecipitation of ORF2g* by P1H1 (Fig 4D–4F), indicating that furin/PC inhibitors abrogated ORF2g maturation. Of note, the cell-permeable CMK and SSM3 inhibitors showed a strong inhibition of ORF2g and αV-pro-integrin maturation, whereas the cell membrane impermeable D6R inhibitor showed a moderate effect on ORF2g maturation. Together, these results indicate that a furin/PC present in the secretory pathway is likely involved in the ORF2g/c maturation process.

Finally, PLC3/HEV-p6 and PLC3 mock cells were transfected with small interfering RNA (siRNA) targeting furin (siFur), or non-targeting siRNA (siCTL) or non-transfected (NT) (Fig 4G). The efficacy of furin silencing was controlled by WB and its effect on ORF2g/c maturation was analyzed as above. As in cells treated with PC inhibitors, we observed an immunoprecipitation of ORF2g* by P1H1 upon furin silencing (Fig 4G), indicating that furin is responsible for ORF2g/c maturation.

## The ORF2 ARM is the regulator of ORF2 addressing

To further analyze the molecular mechanisms by which ORF2 is differentially addressed to the cytosolic, nuclear or reticular pathways, we next generated chimeric and mutant constructs between ORF2 and the CD4 glycoprotein, as a reporter protein. Therefore, we used a heterologous expression system for the further experiments. The constructs were expressed in Huh-7 cells stably expressing the T7 RNA-polymerase (H7-T7-IZ cells) [28]. We selected the 5R/5A and PSG/3R mutations for their marked phenotype (Fig 2) and generated an ARM-deleted mutant (ΔARM, deletion of Gln24 to Arg33, Fig 5). The full-length ORF2wt, ORF2$^{5R/5A}$ and

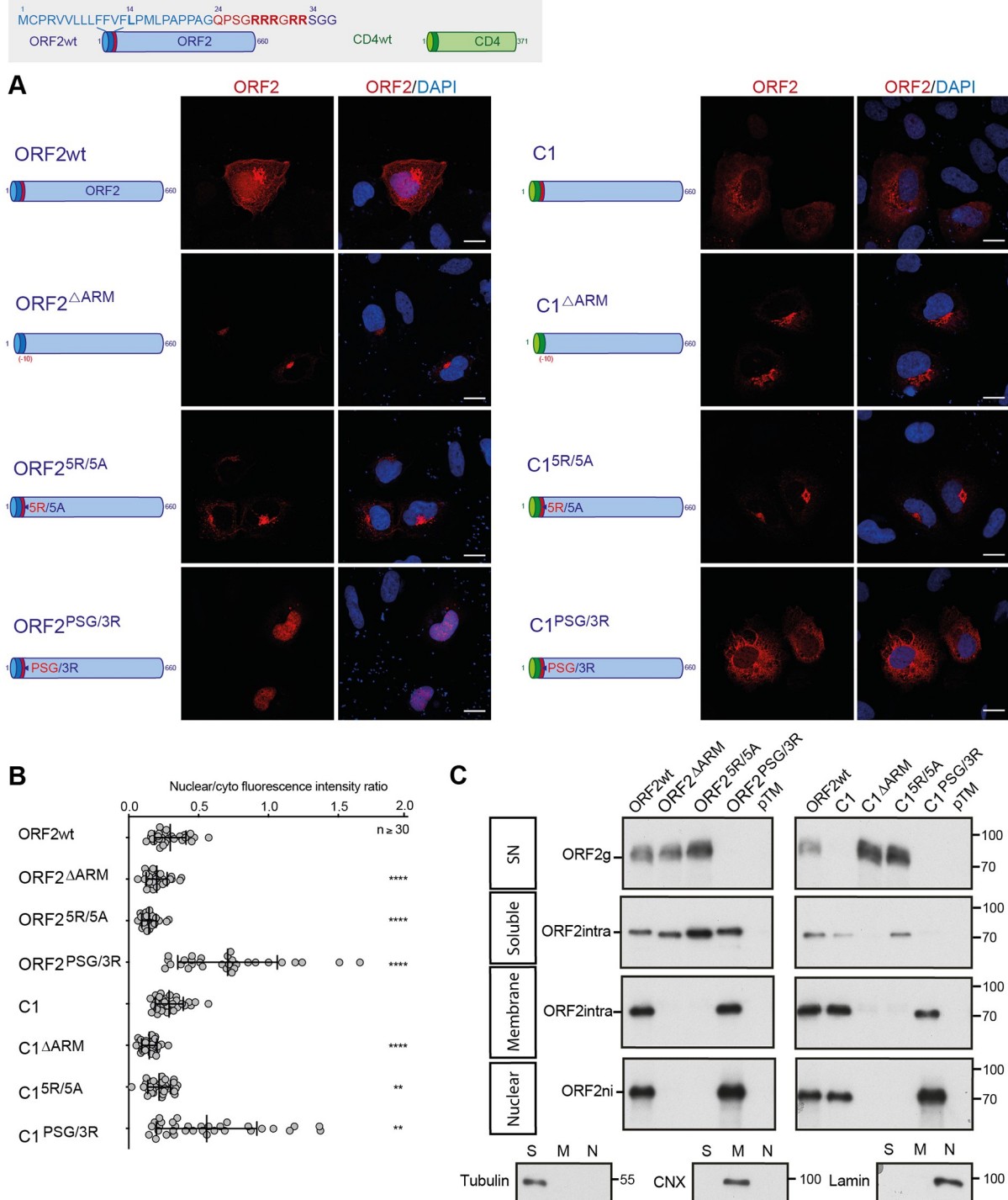

**Fig 5. The ORF2 ARM is the regulator of ORF2 addressing.** Schematic representation of ORF2wt and CD4wt proteins. ORF2 sequences are in blue. ARM residues are highlighted in red. CD4 sequences are in green. (A) H7-T7-IZ cells were transfected with pTM plasmids expressing wt, mutant or chimeric ORF2 proteins. Twenty-four hours post-transfection, cells were fixed and processed for ORF2 staining (in red). Nuclei are in blue. Representative confocal images are shown together with ORF2/DAPI merge images (magnification x63). Blue dots observed in some pictures are DAPI-stained transfected plasmids. A schematic representation of each construct is shown on the left. Scale bar, 20μm. (B) Nuclear-to-cytoplasmic ORF2 staining ratio in H7-T7-IZ cells expressing mutant and chimeric ORF2 proteins. Quantification was done using ImageJ software (mean ± S.D., $n \geq 30$ cells, Kruskal-Wallis with Conover's test). $^{**}p < 0.01$, $^{****}p < 0.0001$. (C) Subcellular fractionation of H7-T7-IZ cells expressing mutant and chimeric ORF2 proteins at 24h post-transfection. Fractionation was done using a subcellular protein fractionation kit for cultured cells. ORF2 proteins were detected by WB with 1E6 Ab. Tubulin, Calnexin (CNX) and Lamin B1 were also detected to control the quality of fractionation. Molecular mass markers are indicated on the right (kDa).

ORF2$^{PSG/3R}$ proteins displayed a similar pattern and phenotype as observed in the infectious system (Fig 2), by immunofluorescence and WB (Fig 5). The effect of the 5R/5A mutation or the ARM deletion on the ORF2 secretion, membrane association and nuclear localization confirmed that the ARM located downstream of the SP negatively regulates ORF2 reticular translocation but is important for nuclear translocation and membrane association. Conversely, the PSG/3R mutation showed an increased nuclear localization and membrane association, whereas ORF2 secretion was fully blocked, confirming that positively charged residues negatively regulate the functionality of ORF2 SP but mediate nuclear translocation and membrane association.

Next, to define the impact of ARM on another SP, we exchanged the ORF2 SP by the one of CD4 that is a functional model SP [29] (Fig 5, Chimeras C1). Interestingly, the chimera C1 displayed a subcellular distribution similar to that of ORF2wt but was no longer secreted (Fig 5C, SN), indicating that the ARM inhibits the functionality of CD4 SP. Indeed, the mutation (C1$^{5R/5A}$) or deletion of ARM (C1$^{\Delta ARM}$) restored secretion of the chimera C1. The observation that ORF2wt is secreted in the presence of the ARM whereas it is not with the CD4 SP, suggests the existence of an interplay between ORF2 SP and ARM. The chimera C1$^{PSG/3R}$ showed an increased nuclear localization and membrane association, whereas its secretion was abolished (Fig 5), confirming that positively charged residues downregulate the functionality of CD4 SP and mediate nuclear translocation and membrane association. It should be noted that the chimera C1$^{PSG/3R}$ showed a marked reticular staining (Fig 5A). As described below, we hypothesized that the CD4 SP might disturb the maturation of ORF2i and anchors the protein into the membrane on the cytosolic side.

We also generated an additional group of ORF2 constructs in which the SP of ORF2 was partially deleted (C2) (S12 Fig). The characterization of these constructs confirmed that ARM mediates ORF2 nuclear translocation and membrane association independently of the reticular translocation.

In order to specifically study the impact of ARM on the functionality of the ORF2 SP independently of the ORF2 ectodomain, CD4 chimeras containing the SP (CD4$^{SPORF2}$), the N-terminus (Chimeras C4) or the ARM of ORF2 (Chimeras C5) were also generated and characterized as previously (Fig 6). Thanks to the CD4$^{SPORF2}$ construct, we confirmed that the SP of ORF2 is a functional SP, as illustrated by its subcellular pattern (Fig 6A) and its efficient secretion (Fig 6D). Interestingly, the chimera C4 showed an intracellular distribution different from that of CD4wt, with a significant nuclear localization (Fig 6A and 6B). In addition, WB analysis revealed a major decrease in C4 secretion as well as the appearance of a 40kDa band in the soluble fraction (Fig 6D). This band corresponds to the non-N-glycosylated CD4 ectodomain, which contains 2 N-glycosylation sites, indicating that the CD4 ectodomain is poorly translocated into the ER lumen when fused to ORF2 N-terminus. The same observations were made for the chimera C4$^{PSG/3R}$. In contrast, the 5R/5A mutations restored the secretion but abolished the nuclear translocation of C4 (C4$^{5R/5A}$, Fig 6A, 6B and 6D). Characterization of the chimeras C5, which contain only the ORF2 ARM, showed results similar to the chimeras C4 (Fig 6B–6D). However, unlike C4, the chimera C5 was no longer secreted (Fig 6D), supporting the hypothesis of an interplay between ORF2 SP and ARM. Moreover, the chimera C5 displayed a reticular staining in addition to nuclear staining (Fig 6C). This observation is in line with our hypothesis that the CD4 SP does not undergo the same maturation as ORF2 SP, and anchors the protein into the membrane with a cytosolic orientation.

Taken together, these results demonstrate that the ORF2 ARM on its own is capable of regulating the functionality of ORF2 or CD4 SP, as well as the nuclear translocation of the protein that carries it.

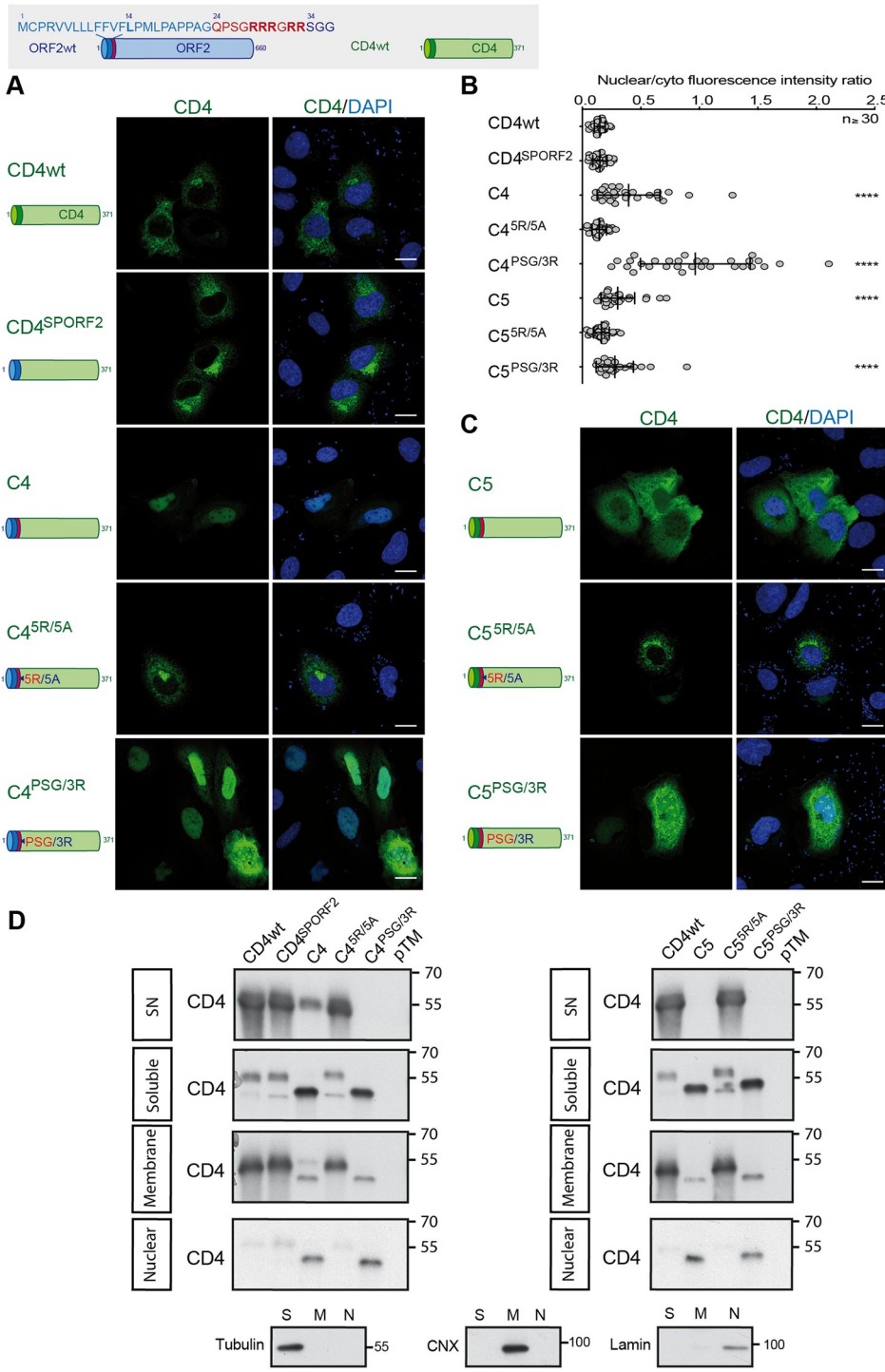

**Fig 6. CD4 addressing by the ORF2 ARM.** Schematic representation of ORF2wt and CD4wt proteins. SP ORF2 residues are shown in blue. ARM residues are highlighted in red. CD4 sequences are in green. (A) and (C), H7-T7-IZ cells were transfected with pTM plasmids expressing CD4wt or chimeric C4 and C5 proteins. Twenty-four hours post-transfection, cells were fixed and processed for CD4 staining (in green). Nuclei are in blue. Representative confocal images are shown together with CD4/DAPI merge images. Blue dots observed in some pictures are DAPI-stained transfected plasmids. A schematic representation of each construct is shown on the left. Scale bar, 20μm. (B) Nuclear-to-cytoplasmic CD4 staining ratio in H7-T7-IZ cells expressing CD4wt or chimeric proteins. Quantification was done using ImageJ software (mean ± S.D., $n \geq 30$ cells, Kruskal-Wallis with Conover's test). $^{****}p < 0.0001$. (D) Subcellular fractionation of H7-T7-IZ cells expressing CD4wt or chimeric proteins at 24h post-transfection. Fractionation was

done using the subcellular protein fractionation kit for cultured cells. CD4 proteins were detected by WB with a rabbit polyclonal anti-CD4 Ab. Tubulin, Calnexin (CNX) and Lamin B1 were also detected to control the quality of fractionation. Molecular mass markers are indicated on the right (kDa).

## The ORF2 ARM regulates the topology of ORF2 SP

Positively charged residues, such as arginine residues, are known to function as determinants of membrane protein topology, which is reflected in a statistical rule of membrane topology, i.e. the positive-inside rule of membrane proteins [30]. Notably, positive charges determine the orientation of the signal sequences and contribute to membrane spanning of the SP H-segment translocating through the translocon [30]. Since our results suggest that the ARM regulates the SP functionality and membrane association of ORF2, we next analyzed the topology of our different ORF2 and CD4 constructs by immunofluorescence (Figs 7 and 8). We used a low concentration of digitonin that selectively permeabilizes the plasma membrane. Triton X-100-permeabilized cells were analyzed in parallel as a control. The differential detection of two epitopes on the ER-membrane associated Calnexin (CNX) was used as a control of permeabilization. We observed that the ORF2wt, the chimera C1 and the PSG/3R mutants displayed a staining in both Triton X-100 and Digitonin-permeabilized cells whereas the ΔARM and 5R/5A constructs showed a labelling only in Triton X-100 permeabilized cells (Fig 7). These accessibility differences in association with the secretion efficiencies (Figs 5C and 6D), which reflect the reticular translocation, allowed us to infer the membrane orientation of each construct as well as the SP topology (Figs 7 and 8). Thus, in the presence of the ARM, the ORF2 SP likely adopts a double topology (ORF2wt) whereas the CD4 SP is not functional (C1) and sticks the protein to the cytosolic side of membranes. The ORF2 SP and CD4 SP are fully functional when the ARM is deleted (ΔARM) or mutated (5R/5A) whereas they are nonfunctional when the ARM is coupled to the PSG/3R mutations. The same observations were done for the C4 and C5 chimeras (Fig 8).

Thus, our findings demonstrate that the ORF2 SP and the ARM act together to direct the fate of ORF2 capsid protein. Thanks to the ARM, the ORF2 SP is likely able to adopt a dual topology leading to either reticular translocation or membrane integration to the cytosolic side.

## Discussion

In the present study, we analyzed a series of mutants of the HEV ORF2 capsid protein to gain insight into how a same primary sequence can generate several ORF2 isoforms with distinctive sequences, post-translational modifications, subcellular localizations and functions in the HEV lifecycle. Several important conclusions can be drawn from our analyses. The first is that the ORF2 protein early transits through the nucleus during infection to control specific cellular functions i.e. antiviral responses of the infected cell. We identified the determinants of the ORF2 nuclear import and export. Notably, an ARM in the N-terminal region of ORF2 mediates nuclear import. Importantly, we showed that the mutation of this motif abolishes ORF2 nuclear translocation but also affects ORF2 addressing into membrane, cytosolic and reticular compartments, which was deleterious for the HEV lifecycle. This brings us to the second important finding, the ARM is pivotal in the fine-tuning of the partitioning and stoichiometry of the ORF2 protein between the nuclear, cytosolic and reticular pathways that are essential to the HEV lifecycle. The last significant finding in this study is the manner by which the SP and ARM cooperate to control the fate of ORF2 protein. Indeed, in addition to mediate the targeting of ORF2 to the ER membrane, the SP is likely able to adopt a reverse signal-anchor

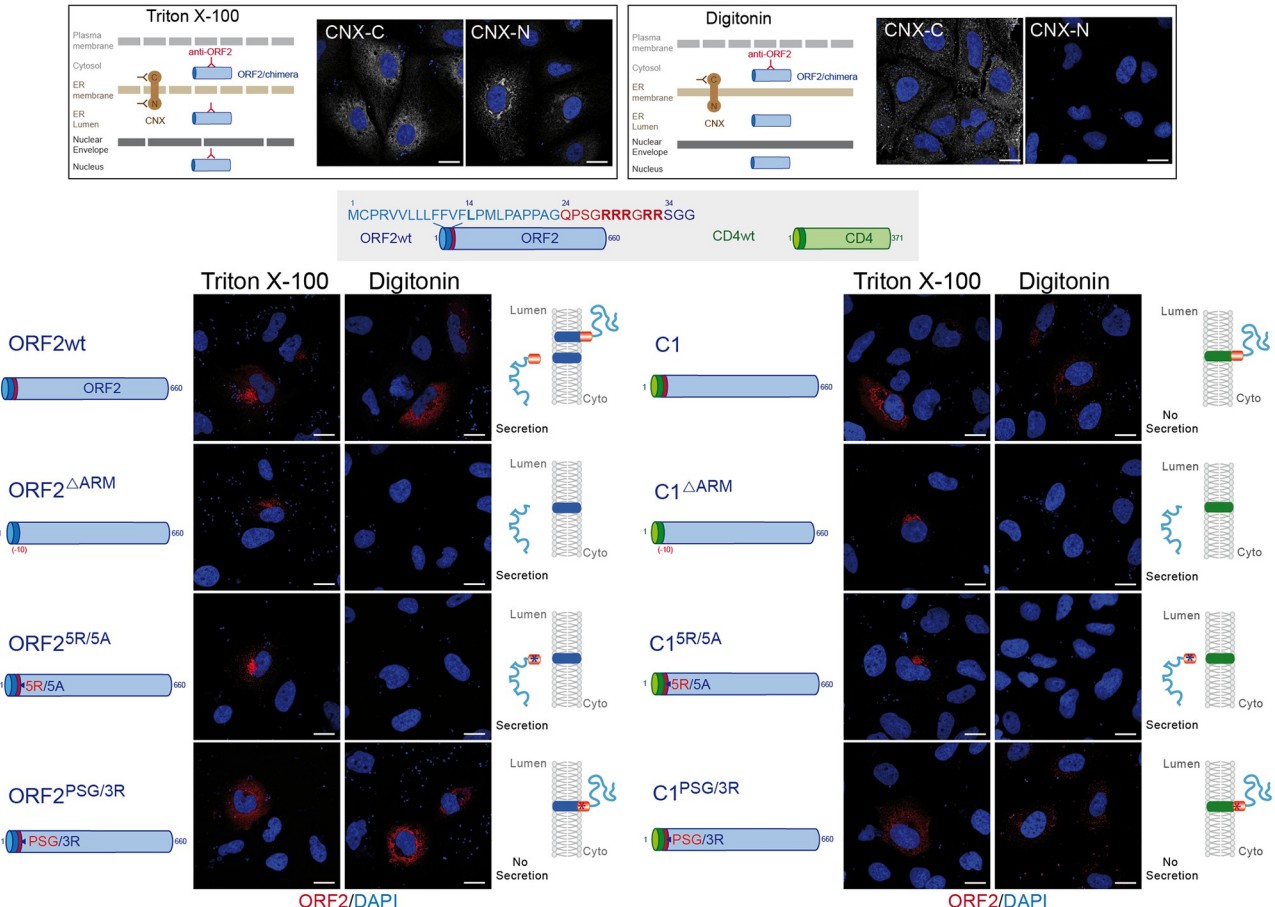

**Fig 7. The ORF2 ARM regulates the topology of ORF2 SP.** A schematic representation of differential permeabilization process with Triton X-100 and Digitonin is shown. Representative images of the differential detection of two epitopes on the ER-membrane associated Calnexin (CNX) used to assess the permeabilization conditions are shown. H7-T7-IZ cells were transfected with pTM plasmids expressing wt, mutant or chimeric ORF2 proteins. Twenty-four hours post-transfection, cells were fixed, permeabilized with either Triton X-100 or Digitonin, and processed for ORF2 staining (in red). Nuclei are in blue. Representative confocal merge ORF2/DAPI images are shown (magnification x63). Blue dots observed in some pictures are DAPI-stained transfected plasmids. A schematic representation of each construct is shown on the left and its predicted topology on the right. Blue and red asterisks correspond to 5R/5A and PSG/3R mutations, respectively. Scale bar, 20μm.

topology. This topology inversion would be driven by flanking charged residues of ARM according to the positive-inside rule [31,32] and leads to the anchoring of the ORF2i protein to the cytosolic side of membranes.

Previously, we [12] and others [15] demonstrated that HEV produces several isoforms of the ORF2 capsid protein. The ORF2i protein is the structural component of infectious particles. It is likely derived from the assembly of the ORF2intra form present in the cytosolic compartment. The ORF2i and ORF2intra proteins are not glycosylated and display the same sequence starting at Leu14 corresponding to the middle of the SP, indicating that an intramembrane protease might be involved in their maturation. Further investigation is required to identify this intramembrane protease. In contrast, ORF2g/c proteins are highly glycosylated and secreted, but are not associated with infectious material. We identified the first residues of ORF2g/c as Ser34 and Ser102, respectively [12,13]. The nature of the sequences upstream of the ORF2g/c N-termini, experiments using PC inhibitors and siRNA transfection indicate that Furin is involved in their maturation [33]. Interestingly, the ARM is right upstream of the N-

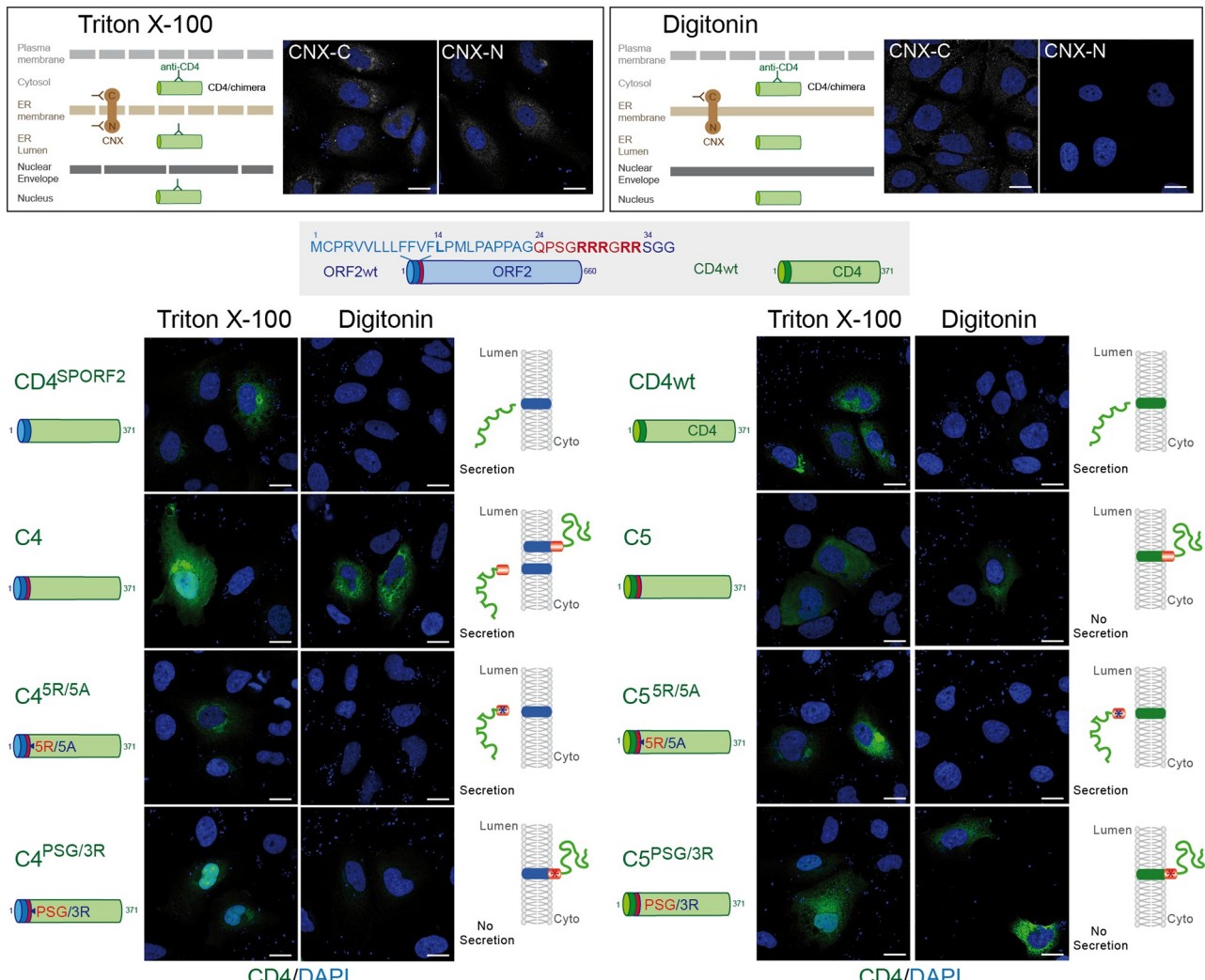

**Fig 8. The ORF2 ARM is also able to control the topology of CD4 SP.** A schematic representation of differential permeabilization process with Triton X-100 and Digitonin is shown. Representative images of the differential detection of two epitopes on the ER-membrane associated Calnexin (CNX) in the used permeabilization conditions are shown. H7-T7-IZ cells were transfected with pTM plasmids expressing CD4wt or chimeric C4 and C5 proteins. Twenty-four hours post-transfection, cells were fixed, permeabilized with either Triton X-100 or Digitonin, and processed for CD4 staining (in green). Nuclei are in blue. Representative confocal CD4/DAPI merge images are shown. Blue dots observed in some pictures are DAPI-stained transfected plasmids. A schematic representation of each construct is shown on the left and its predicted topology on the right. Blue and red asterisks correspond to 5R/5A and PSG/3R mutations, respectively. Scale bar, 20μm.

terminus of ORF2g, indicating that this motif also serves as a recognition site for ORF2g maturation.

We demonstrated that early during replication and infection, the ORF2 protein from different strains transits through the nucleus likely to control antiviral responses of the infected cell. Although we identified the determinants of nuclear import and export in the ORF2 sequence, further studies are now required to identify the cellular partners of nuclear ORF2 and define precisely the impact of ORF2 host regulation on immune cell responses. A previous study demonstrated the significance of ORF2 N-terminal arginine residues in the inhibition of TBK1-mediated IRF3 phosphorylation [34]. Although they did not take into account the effect of arginine residues on ORF2 partitioning, their results are in line with our observations on ORF2 interference with the host innate immunity.

Based on our findings, we propose a model of ORF2 production. Firstly, when engaged with the translocon, the ORF2 SP initially inserts head-on in an $N_{exo}/C_{cyt}$ orientation (Fig 9A). Two mechanisms can then take place. In one side, ORF2 SP inverts orientation to $N_{cyt}/C_{exo}$ to integrate the ER membrane as cleavable signal. The C-terminal end of signal is exposed to the

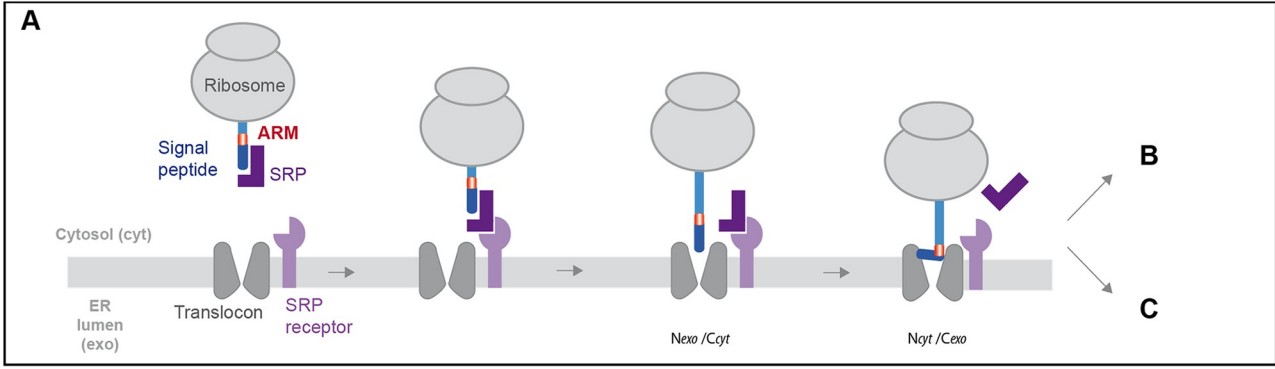

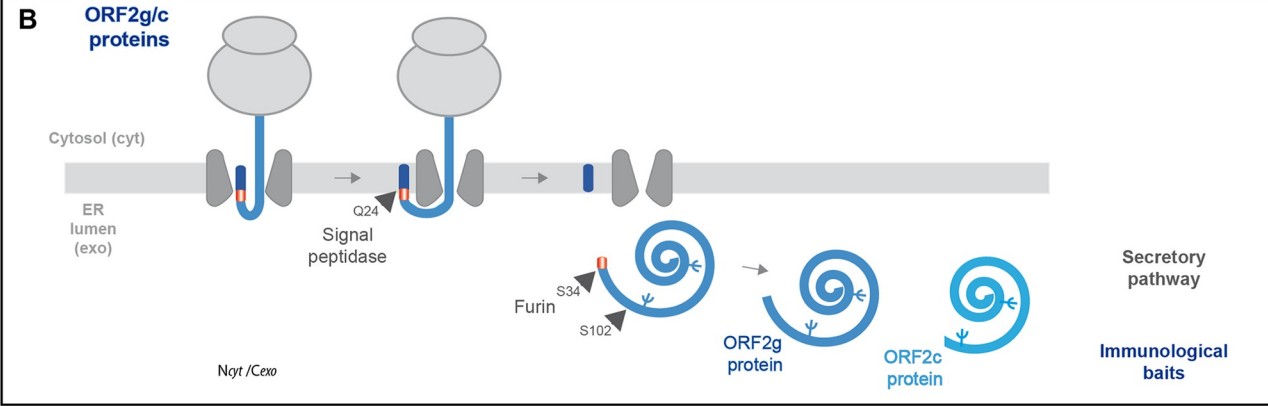

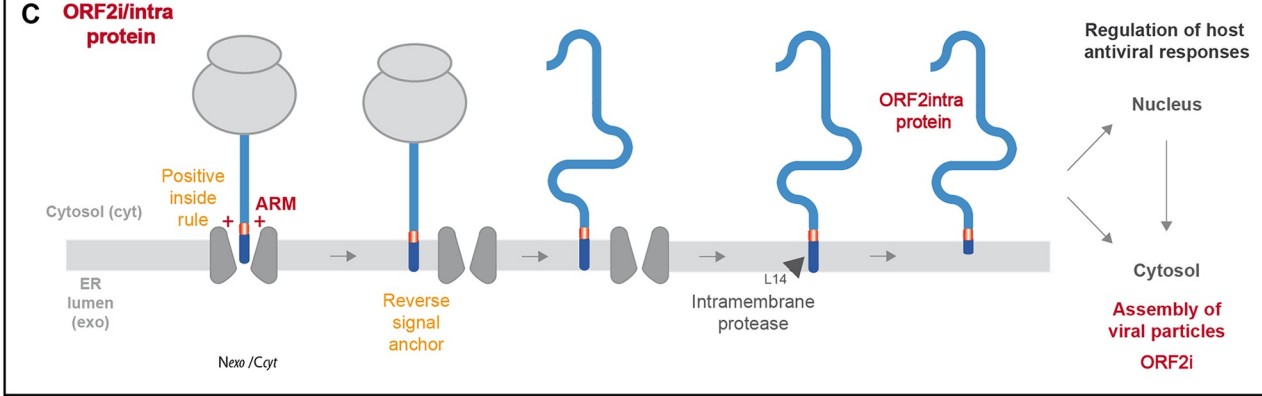

**Fig 9. Model of ORF2 addressing regulation by ARM.** (A) The signal recognition particle (SRP) recognizes the hydrophobic signal peptide (SP) of the ORF2 nascent chain as it emerges from a translating ribosome. The ribosome-nascent chain-SRP complex is targeted to the membrane and interacts with the SRP receptor, resulting in the release of the SP and docking of the ribosome–nascent chain complex to the Sec61 translocon. The ORF2 SP initially inserts head-on in an $N_{exo}/C_{cyt}$ orientation, then inverts its orientation to $N_{cyt}/C_{exo}$. (B) The C-terminal end of SP is exposed to ER lumen and is cleaved by signal peptidase, generating a new N-terminus. Translation then resumes, and the nascent ORF2 protein is translocated into the ER lumen where it is glycosylated and likely undergoes maturation by the proprotein convertase furin. This pathway generates the ORF2g/c forms. (C) For a fraction of ORF2 nascent polypeptide chains, the ARM leads the ORF2 SP to retain its $N_{exo}/C_{cyt}$ orientation and integrates as reverse signal-anchor, according to the positive-inside rule. The ORF2 protein anchored to the cytosolic side of membrane is likely processed by an intramembrane protease to generate the ORF2i/ORF2 intra protein that is translocated into the nucleus and assembles into viral particles.

ER lumen and cleaved by signal peptidase, liberating the ORF2 ectodomain in the ER lumen where it undergoes glycosylation and protease maturation. This pathway leads to the production of ORF2g/c proteins that are abundantly secreted and likely serve as immunological baits (Fig 9B). On the other side, ORF2 SP does not invert, keeps a $N_{exo}/C_{cyt}$ orientation and serves as a reverse signal anchor. Next, the ORF2 protein anchored to the cytosolic side of membranes is likely processed by an intramembrane protease (Fig 9C). This pathway leads to the production of the ORF2intra protein that is early translocated into the nucleus to play immunomodulatory functions and/or is then assembled into viral particles in the cytosolic compartment. Of note, the dual topology was exclusively observed for the ORF2 and CD4 constructs containing both ORF2 SP and ARM (ORF2wt and chimera C4), reflecting the specific interplay between ORF2 SP and ARM and no involvement of other sequence determinant in this process.

Due to the size constraint of their extracellular phase, viruses are under strong pressure to minimize the size of their genome. Overlapping genes represent an adaptive strategy developed by many viruses to condense a maximum amount of information into short nucleotide sequences. This gene overlap strategy is used by HEV for the ORF2 and ORF3 expression [35] and ORF4 in gt1 [36]. Here, our study enabled the pioneering demonstration that HEV also developed a master strategy to condense multiple information into only five amino acid residues of its capsid protein. This strategy is likely also exploited by other pathogens. Thus, the ORF2 ARM is a central regulator of the HEV lifecycle that controls (i) the ORF2 nuclear localization and hereby controls cellular functions promoting regulation of host antiviral responses, (ii) the functionality of ORF2 SP leading to the production of either viral particles or humoral immune decoys, (iii) maturation of glycosylated ORF2 protein by furin, and (iv) membrane association that is likely essential to particle assembly. Hence, ARM controls the fate and function of ORF2 capsid protein isoforms and is likely a key player in the hijacking of cellular processes.

## Materials and methods

### Cell cultures

PLC3 [12] and Huh-7.5 [37] cells were cultured as previously described [13]. The Huh-7-derived H7-T7-IZ cells stably expressing the T7 RNA polymerase ([28]; kindly provided by Ralf Bartenschlager, University of Heidelberg, Germany) were maintained in a medium supplemented with 50 µg/ml of Zeocin. They were used for the transfection of the T7 promoter-driven pTM expression vectors.

### Plasmids and transfection

The plasmid pBlueScript SK(+) carrying the DNA of the full-length genome of adapted gt3 Kernow C-1 p6 strain, (GenBank accession number JQ679013, kindly provided by S.U Emerson) was used as a template [11]. Mutants of the ORF2 ARM or NES sites were generated by site directed mutagenesis. Individual mutations were introduced by sequential PCR steps, as described previously [13], using the Q5 High-Fidelity 2X Master Mix (New England Biolabs, NEB), then digestions with restriction enzymes and ligation were performed. All the mutations were verified by DNA sequencing. The primers used for the generation of ORF2 mutants are listed in S1 Table. The ORF3-null mutant of HEV-p6 (HEV-p6-ΔORF3) was generated as described in [38].

The plasmid pSK-HEV-2 carrying the DNA of the full-length genome of gt1 Sar55 strain (GenBank accession number AF444002) was kindly provided by X-J. Meng. The plasmid

encoding the full-length gt3 HEV83-2 clone (GenBank accession number AB740232) was kindly provided by K. Ishii as well as T. Wakita.

Capped genomic HEV RNAs were prepared with the mMESSAGE mMACHINE kit (Ambion) and delivered to PLC3 cells by electroporation using a Gene Pulser Xcell apparatus (Bio-Rad) [12].

The plasmids pTM-ORF2 (kindly provided by J. Gouttenoire, University of Lausanne, Switzerland) [14] and pTM/CD4 have been previously described [14,39]. The pTM/CD4 contains the DNA sequence coding for the secreted ectodomain of CD4 (aa 1–371). The primers used for the generation of ORF2/CD4 chimeras/mutants are listed in S1 Table. For some constructs, PCR amplifications were performed by multiple heat pulses [40]. The pTM plasmids were transfected into H7-T7-IZ cells using ViaFect Transfection Reagent (Promega) following the manufacturer's recommendations.

## Antibodies

Primary antibodies used in this study are listed in Table 1. Secondary antibodies were from Jackson ImmunoResearch.

## Indirect immunofluorescence

Immunostainings were performed as previously described [13]. For selective permeabilization experiments, cells were fixed with 2% of PFA, washed twice with PBS and permeabilized for 30 min at 4°C with either 0.01% of Digitonin (Sigma) in buffer containing 20 mM HEPES pH6.9, 0.3 M sucrose, 0.1 M KCl, 2.5 mM MgCl2 and 1 mM EDTA or 0.5% Triton X-100 in PBS. Cells were next stained with buffers containing Digitonin.

The nuclei were stained with DAPI and cell outlines with CellMask Green (Invitrogen). Cells were analyzed with a LSM 880 confocal laser-scanning microscope (Zeiss) using Plan Apochromat 63xOil/1.4N.A. and EC Plan Neofluar 40xOil/1.4N.A. objectives. The images were processed using ImageJ software.

**Table 1. Primary antibodies used for Western-blot, immunoprecipitation and indirect immunofluorescence experiments.**

| Target | Clone / Species / Isotype | Reference | Supplier | Antibody registry reference |
|---|---|---|---|---|
| HEV ORF2 | 1E6 / Mouse / IgG2b | MAB8002 | Millipore | AB-827236 |
| HEV ORF2 | 2E2 / Mouse / IgG1 | MAB8003 | Millipore | AB-571017 |
| HEV ORF2 | 4B2 / Mouse / IgG1 | MAB8006 | Millipore | AB-11212267 |
| HEV ORF2 | P1H1 / Mouse / IgG3 | Patent # PCT/EP2019/068338 [41] | Home-made | x |
| HEV ORF2 | P3H2 / Mouse / IgG3 | [41] | Home-made | x |
| HEV ORF3 | Rabbit | Graff et al. [38] | S. Emerson (NIH) | x |
| CD4 | Rabbit | ab133616 | Abcam | AB-270883 |
| Importin α-1 | Goat | ab6036 | Abcam | AB-30524 |
| Importin α-1 | Rabbit | PA5-83544 | Invitrogen | AB-2790697 |
| CRM1 | Rabbit | #46249 | Cell Signaling | AB-2799298 |
| γ-Tubulin | Mouse / IgG1 | #T-5326 | Sigma-Aldrich | AB-532292 |
| CNX-N | Rabbit mAb | ab92573 | Abcam | AB-10563673 |
| CNX-C | Rabbit | ab22595 | Abcam | AB-2069006 |
| SP1 | Rabbit | #PA5-29165 | Thermofisher | AB-2546641 |
| GM130 | Mouse / IgG1 | #610822 | BD Biosciences | AB-398141 |
| Lamin B1 | Rabbit | ab16048 | Abcam | AB-443298 |
| αV-integrin | Rabbit | ab124968 | Abcam | AB-11129746 |
| Furin | MON-152 / Mouse / IgG1 | ALX-803-017-R100 | Enzo Life Sciences | AB-2051436 |

## Quantification of nuclear/cytosolic fluorescence intensity ratios

Cells were co-stained with CellMask Green (Invitrogen) and analyzed using ImageJ software with a method adapted from McCloy *et al.* [42]. The regions of interest (ROI) were drawn around the whole cells and the nuclei. Area, integrated density and mean gray values were measured. For each cell, the nuclear/cytosolic fluorescence intensity ratio was calculated by the following formula: exact fluorescence intensity of nucleus / (exact fluorescence intensity of whole cell—exact fluorescence intensity of nucleus).

The exact fluorescence was = the corrected total cell fluorescence (CTCF)—the mean of the integrated density of non-infected cells. CTCF was calculated by the following formula: CTCF = integrated density–(area of selected electroporated cells x mean of background fluorescence around the cells).

## Pearson's correlation coefficient (PCC) determination

Colocalization studies were performed by calculating the PCC using the JACoP plugin of ImageJ software. For each calculation, at least 30 cells were analyzed.

## Virus production and intracellular viral particles preparation

PLC3 cells were electroporated with HEV-p6 RNAs as previously described [12]. Intracellular viral particles were prepared by osmotic shock, as previously described [13].

## Infectious titer determination

Infectious titers in focus forming unit (FFU/mL) were determined as described previously [13] by using Huh-7.5 cells. ORF2-positive cells were quantified using an InCell 6000 confocal analyzer (GE Healthcare) and the Columbus image analysis software (Perkin Elmer).

## Chemicals and viability assay

Leptomycin B (Cell Signaling), Verdinexor (AdooQ Biosciences), Gossypol (Tocris), Mycolactone A/B toxin [43], Decanoyl-RVKR-chloromethylketone [CMK] (Sigma), hexa-D-arginine amide [D6R] (Sigma) and SSM3 trifluoroacetate (Tocris) were used in this study. Dose-response curves of PLC3 cells treated with the different drugs are shown in S13 Fig. Cell viability was assessed using a CellTiter 96 AQueous Non-Radioactive Cell Proliferation Assay (Promega).

## RNA extraction and quantification

HEV RNA and cellular gene RNA levels were quantified by RT-qPCR as described in S1 Text.

## Western blotting analysis

Western blotting analyses were performed as described previously [12]. Target proteins were detected with specific antibodies (Table 1) and corresponding peroxidase-conjugated secondary antibodies.

## Immunoprecipitations

Anti-ORF2 antibodies were bound to magnetic Dynabeads M-270 Epoxy beads (Thermofisher) overnight at 37˚C following the manufacturer's recommendations. Beads were washed and then incubated for 1h at RT with cell lysates or heat-inactivated supernatants. Beads were

washed and then heated at 80˚C for 20 min in Laemmli buffer. Proteins were separated by SDS-PAGE and ORF2 proteins were detected by WB using the 1E6 MAb.

## Proximity ligation assay (PLA)

Electroporated PLC3 cells cultured on glass coverslips were fixed with 3% paraformaldehyde for 20 min, methanol for 5min and permeabilized in PBS containing 0.1% Triton X-100 for 30 min. Proximity ligation assay was performed using Duolink *in situ* detection kit (Sigma), as recommended by the manufacturer, with mouse anti-ORF2 1E6 MAb and rabbit anti-Importin-α1 MAb or rabbit anti-CRM1 MAb (Table 1). Images were acquired by confocal microscopy, as described above. For each condition, twelve fields were acquired. For each field, a stack of images corresponding to the total volume of the cells was acquired. Maximum-intensity projection images were generated using Zen software. Representative images were assembled and dots were counted using ImageJ software. For each field, the mean number of spots per cell was calculated by dividing the number of spots by the number of nuclei.

## Silencing experiments

PLC3/HEV-p6 and PLC3 mock cells were transfected with small interfering RNA (siRNA) pools (Horizon) targeting furin (ON-TARGETplus human Furin, gene 5045, siRNA SMART-pool) or with a nontargeting siRNA control (siCTL) by using RNAiMax reagent (Invitrogen) according to the manufacturer's instructions. The knockdown effects were determined at 72h post-transfection by western-blotting and immunoprecipitation.

## Subcellular extraction

Extractions were performed with the Subcellular Protein Fractionation Kit for Cultured Cells (Thermo scientific) following the manufacturer's recommendations. The cytoplasmic extracts were ultra-centrifuged at 100 000 g during 1h at 4˚C. Anti-β-tubulin (cytoplasmic), anti-Calnexin (membranous) and anti-SP1 or anti-Lamin B1 (nuclear soluble) antibodies were used to control the quality of extractions.

## Statistical analyses

Statistical analyses were performed with the software RStudio version 1.2.5001 combined with R version 3.6.1. For all statistical tests, reported p values were two-sided. A test was declared statistically significant for any p value below 0.05. For comparing more than three groups of unpaired data, ANOVA or its non-parametric equivalent test, the Kruskal-Wallis test, was used. ANOVA was preferred when the distributions in each group followed a normal distribution, and the assumption of equality of the variances between each group was verified. When tests showed a significant difference between the groups, post hoc tests were performed. The Dunnett test (dunnettTest function with default option) followed an ANOVA, and the Conover's test (kwManyOneConoverTest function with pvalues adjusted by the Benjamini-Hochberg procedure) followed the Kruskal-Wallis test. Each group was compared to a reference. For the kinetic experiments, the Friedman test was preferred because of a lack of normality and homogeneity of variance and the post hoc Nemenyi test (frdManyOneNemenyiTest function from the package PMCMRplus) was used. For comparing two groups with unpaired data the Student's t test was used when gaussian distributions and homogeneity of variances were observed. If variances between groups were different, the Welch's correction was applied to the t test. And in case of a lack of normality of data, the Mann-Whitney's U test was chosen.

## Supporting information

**S1 Text. Legends of supporting figures, supporting materials and methods, and supporting references.**
(DOCX)

**S1 Fig. Conservation of the Arginine-Rich Motif (ARM) in the ORF2 sequence.**
(TIF)

**S2 Fig. Effect of 5R/5A, PSG/3R and ΔSP mutations on the kinetics of ORF2 subcellular localization.**
(TIF)

**S3 Fig. Colocalization analysis of ORF2 with the Golgi marker 130 (GM130) in PLC3/HEV-p6 cells expressing ORF2wt or ARM/SP mutants.**
(TIF)

**S4 Fig. Controls of RNA and infectious titers**
(TIF)

**S5 Fig. Colocalization analysis of ORF2 with the Importin-α1 in PLC3/HEV-p6 cells expressing ORF2wt and ARM/SP mutants.**
(TIF)

**S6 Fig. ORF2 nuclear translocation modulates host gene expression.**
(TIF)

**S7 Fig. Effect of ΔORF3 and NES12 mutations on the kinetics of ORF2 subcellular localization.**
(TIF)

**S8 Fig. Nuclear export, colocalization and interaction of ORF2 with the exportin CRM1 in cells expressing ORF2wt or NES mutants.**
(TIF)

**S9 Fig. Conservation of the Nuclear Export Signal 9 (NES9) in the ORF2 sequence.**
(TIF)

**S10 Fig. Conservation of the Nuclear Export Signal 10 (NES10) in the ORF2 sequence.**
(TIF)

**S11 Fig. Conservation of the Nuclear Export Signal 12 (NES12) in the ORF2 sequence.**
(TIF)

**S12 Fig. Addressing of C2 constructs.**
(TIF)

**S13 Fig. Dose-response curves of PLC3 cells treated with the different drugs used in this study.**
(TIF)

**S1 Table. Primers of use to generate ORF2/CD4 chimeras/mutants.**
(DOCX)

**S2 Table. Primers used for RT-qPCR of cellular genes.**
(DOCX)

**S1 Data. Excel spreadsheet containing, in separate sheets, the underlying numerical data and statistical analysis for Figs 1 and 2B, 2D Top, 2D Bottom and 2E, 3A, 3C, 3E and 3F, 5B and 6B, S2, S3, S4, S5, S6A Left, S6A Right, S6D, S7, S8A, S8B, S8C, S12, and S13 Figs.** (XLSX)

**S2 Data. Compressed file containing figures of the unprocessed gels used in Figs 2C, 3D, 4A, 4C, 4D–4F and 4G, 5C and 6D, S7A and S12C Figs.** (ZIP)

## Acknowledgments

We thank Olivia Beseme for her technical contribution. We thank Suzanne U. Emerson (NIH, USA), Jérôme Gouttenoire (University of Lausanne) and Ralph Bartenschlager (University of Heidelberg) for providing us with reagents. We thank François-Loïc Cosset (University of Lyon) for critical reading of the manuscript.

## Author Contributions

**Conceptualization:** Kévin Hervouet, Martin Ferrié, Maliki Ankavay, David Hot, Jean Dubuisson, Cécile-Marie Aliouat-Denis, Laurence Cocquerel.

**Data curation:** Laurence Cocquerel.

**Formal analysis:** Kévin Hervouet, Martin Ferrié, Maliki Ankavay, Cécile Lecoeur, David Hot, Thibaut Vausselin, Laurence Cocquerel.

**Funding acquisition:** Laurence Cocquerel.

**Investigation:** Kévin Hervouet, Martin Ferrié, Maliki Ankavay, Claire Montpellier, Charline Camuzet, Virginie Alexandre, Aïcha Dembélé, Arnold Thomas Foe, Peggy Bouquet, Jean-Michel Saliou.

**Methodology:** Kévin Hervouet, Martin Ferrié, Maliki Ankavay, Claire Montpellier, Laurence Cocquerel.

**Project administration:** Laurence Cocquerel.

**Resources:** Sophie Salomé-Desnoulez, Alexandre Vandeputte, Laurent Marsollier, Priscille Brodin, Marlène Dreux, Yves Rouillé.

**Supervision:** Laurence Cocquerel.

**Validation:** Kévin Hervouet, Martin Ferrié, Maliki Ankavay, Claire Montpellier, Cécile Lecoeur, Thibaut Vausselin, Laurence Cocquerel.

**Visualization:** Laurence Cocquerel.

**Writing – original draft:** Laurence Cocquerel.

**Writing – review & editing:** Cécile-Marie Aliouat-Denis, Laurence Cocquerel.

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
