## [Decision Letter · Decision Letter 0]

2 Dec 2021

Dear Dr. Cocquerel,

Thank you very much for submitting your manuscript "An Arginine-Rich Motif in the ORF2 Capsid Protein Regulates the Hepatitis E Virus Lifecycle and Interactions with the Host Cell" for consideration at PLOS Pathogens. As with all papers reviewed by the journal, your manuscript was reviewed by members of the editorial board and by several independent reviewers. In light of the reviews (below this email), we would like to invite the resubmission of a significantly-revised version that takes into account the reviewers' comments.

We cannot make any decision about publication until we have seen the revised manuscript and your response to the reviewers' comments. Your revised manuscript is also likely to be sent to reviewers for further evaluation.

Sincerely,

Alexander Ploss, Ph.D.

Guest Editor

PLOS Pathogens

Guangxiang Luo

Section Editor

PLOS Pathogens

Kasturi Haldar

Editor-in-Chief

PLOS Pathogens

orcid.org/0000-0001-5065-158X

Michael Malim

Editor-in-Chief

PLOS Pathogens

orcid.org/0000-0002-7699-2064

Reviewer's Responses to Questions

**Part I - Summary**

Reviewer #1: The life cycle of HEV is poorly understood due to lack of robust cell culture model. ORF2 is the viral capsid protein, which could package viral genome and mediate viral entry by engagement with the putative cellular receptor. Interestingly, ORF2 has multiple forms and they play different roles in HEV life cycle. In this manuscript, Hervouet et al constructed a panel of ORF2 mutants in Arginine-Rich Motif (ARM) to delineate the function of ARM motif in regulating ORF2s subcellular localization and host antiviral response. ARM regulates the production of different forms of ORF2 and serves as maturation site of glycosylated ORF2. This study could further deep our understanding of HEV life cycle and its interaction with cell host, however, this manuscript is not well organized, not necessarily long and the experiments are not well-control, other explanations or conclusions can be drawn. In addition, the authors performed lots of experiments by electroporation of viral RNA, instead of viral infection, which could potentially lead to some artifacts of the observation. The specific comments could be found as below.

Reviewer #2: In their manuscript entitled “An Arginine-Rich Motif in the ORF2 Capsid Protein Regulates the Hepatitis E Virus Lifecycle and Interactions with the Host Cell” Hervouet et al. use an HEV cell culture model to elucidate the role of ARM on multiple steps of HEV propagation.

Utilizing a series of ORF2 mutants expressed either in the p6 HEV strain or as ORF2 capsid protein they delineate a crucial role of the ARM as functional nuclear localization signal which seems to be important for viral production. With the help of specific inhibitors and co-localization studies, the authors identified a CRM1-dependent egress from the nucleus and 3 important nuclear export signals within the ORF2 protein.

The main body of the manuscript focusses on the maturation and addressing of ORF2. The authors found evidence for a role of a furin-dependent secretion pathway in the maturation process. With the generation and evaluation of additional mutants including the signal and/or the encoding sequence of CD4 by state-of the art techniques the authors delineate a tight interplay of the ARM with the respective signal peptide for addressing, topology and egress of the capsid proteins. Bringing this data set together the authors finally delineate a model for the generation of the infectious or non-infectious forms of the HEV particles.

The manuscript is well written, the experiments are appropriate and on a technically high level. Following on the previous reports from the group this manuscript sheds light into the so far under investigated HEV life-cycle and the generation of the different capsid forms.

Overall, this manuscript describes several novelties in the field and sets the ground for several future scientific perspectives concerning HEV.

**Part II – Major Issues: Key Experiments Required for Acceptance**

Reviewer #1: 1) P6 L114 why the authors choose electroporation of viral RNA genome to observe the ORF2 localization, instead of viral infection? A large amount of viral RNA delivered into the cells by electroporation could produce lots of ORF2, which is not an appropriate model to observe ORF2 localization.

2) The authors constructed series of ARM mutants to observe the alterations of ORF2 mutants localization to demonstrate ARM’s NLS function. Also they engineered the mutations accordingly to demonstrate the critical role of ORF2 nuclei localization in HEV life cycle (Fig 2A-D). It has been well known that ORF2 is also responsible to package viral genome to form infectious particles, and it is very possible that the ARM mutation impaired ORF2 function to package viral genome, which is not related to its NLS function. I suspect these mutations affect ORF2 fold or interaction with genomic RNA to cause the phenotype.

3) In addition, the authors also need to check the total ORF2 amount in Fig 2C, excluding the possibility that the mutations in ARM affect ORF2 stability. The cellular markers in the WB were required to make sure the quality of isolation of each fraction. For example, there is not nuclei maker (histone protein) in the nuclei extract, which is very important.

4) Data presented in Figure 2D show the ORF2 mutants produce less infectious viral particles than WT. As the ORF2 and ORF3 are overlapped, in ORF2 mutants, the amino acid of ORF3 also changed, therefore, more data should be presented to exclude the possibility that this phenotype is caused by ORF3 mutation.

5) The authors used multiple inhibitors to block the interested pathway and then analyze the phenotype (Fig 3 and Fig 4), and always have the concerns about the specificity of the phenotype in the manuscript. More solid data should be used to support the conclusions.

Reviewer #2: I have the following major comments.

Major:

All experiments have been performed with the (sub-)genotype Kernow p6. As the authors also stated in their introduction, several other (sub-)genotypes exist with marked differences in their clinical presentation. Therefore, this study would largely benefit from expanding the range of viruses used. In this regard it would be helpful to

I) show by phylogenetic analysis of for example published strains the abundance of ARM as well as NES

II) show a similar nuclear pattern for a different GT3 subgenotype as well as another genotype (the published data so far only investigated GT3 infections).

The main body of the manuscript is focussing on virus addressing. Some parts are comparable little studied and therefore do not allow as strong statements as have been made in the manuscript. This in particular is the case for the following:

I) Page 7, Lines 140-143: “In addition, high molecular weight forms

of ORF2intra in the soluble fraction as well as an increase of ORF2g/c secretion were observed (Fig2C), suggesting a higher translocation into the secretory pathway associated to an improved functionality of ORF2 SP”

This statement seems only to be true for 3R/3A and 5R/5A, not for the 2R/2A construct. The authors should either repeat the experiments to identify significant results or be gentle with the description.

II) Page 7, Lines 166-168:”Intracellular titers were also

lowered in SP mutants (Fig2D), indicating that the ORF2 SP likely plays an important role in the assembly of infectious particles.”

The intracellular HEV RNA is already significantly lower for the SP mutants intracellularly and therefore argue for a reduced replication. Overall, this figure would benefit from a positive control (such as ribavirin) and possibly a time-course to ensure a decent measuring window.

III) Page 8, Lines 177-179 “Interestingly, ORF2 and Importin-α1 co-localized in the nucleus of infected cells with a Pearson correlation coefficient (PCC) of 0.670, and the mutation of arginine residues drastically reduced this colocalization. Taken together, our results indicate that ORF2 colocalizes with Importin-α1 thanks to its ARM that serves as a functional NLS.”

This seems to be only a moderate co-localization. More work needs to be done to support this assumption such as performing IP experiments to reveal direct interaction.

IV) Page 13, Lines 203-207: “Altogether our results suggest that the ORF2 ARM, which notably regulates ORF2 nuclear translocation, is a pivotal viral determinant for the modulation of host pathways and, especially, genes of the NF-κB-induced signalling upon infection. Further studies will be required to define precisely the impact of this HEV-driven host regulation on immune cell responses.”

Although it seems to be an interesting observation, I agree that the displayed data is too preliminary and should be left out or expanded significantly (using different genotypes of HEV, display all identified hits). Additionally, the time-course of viral replication might be significantly affected depending on whether the construct enters the nucleus or not. Therefore, the analysis at one specific time-point (18h) is difficult to compare. Additionally, mutant delORF3 has not been investigated in the previous experiments in terms of translocation to the nucleus.

V) Fig 7/8. Have the authors analysed an additional SP from a different source in terms of its phenotype?

**Part III – Minor Issues: Editorial and Data Presentation Modifications**

Reviewer #1: 1) Fig 1, it is required to present the nuclei staining results.

2) Fig 2D needs RdRp inactive (GAD) control.

3) P9 L187 whether overexpression of ORF2 can cause similar results on NF-kB-related signaling.

4) P10 L230 The phenotype is not consistent among these three mutants, suggesting that there is nothing to do with nuclear localization.

5) P13 L287-L289 The localization of ORF2 is also dynamic like infection condition? If not, OE system cannot reflect true functional ORF2 in infection condition.

6) Figure 2C. Tubulin of ΔSP1 is much less than the others, indicating the quality of ΔSP1 soluble extract is not good.

7) Figure 4A and 4D-F. At What time points post-electroporation are the cells treated with inhibitors? Some related descriptions should be added to the legend.

Reviewer #2: Overall, the study would benefit from additional controls and more detailed analysis in particular in the first part.

1) Supplement Figure 6: Please provide data with addition of a positive/cytotoxic control.

2) Page 7 Lines 151-155: he PSG/3R mutant showed

a marked nuclear localization (Figs2Band 2C, Nuclear extract) but was impaired in

ORF2g/c secretion (Fig2C, Supernatant), as observed for the SP deletion mutants,

indicating that the addition of arginine residues strengthens the NLS function of ARM

but inhibits the functionality of ORF2 SP.

To underline this point it would be interesting to perform an additional time-kinetic (such as in Fig1) to understand and learn where the ORF2 dose eventually reside and if it is retained in the nucleus for a longer period of time.

3) Page 14, Lines 231-234, “Of note, intracellular replication was

not altered by NES mutations (Fig3F), indicating that the loss of infectious particle

assembly is due to the differential subcellular localization of the mutants and not to a

replication defect.”

This seems counterintuitive. Please provide appropriate replication inhibitor control of the assay. Additionally, this would also benefit from a time-kinetic analysis.

4) In line 114 page 6: Titel: “phase of infection”: This is not correct, as the cells were transfected. Therefore, the term replication would be more appropiate.

5) Fig 5C convincingly shows that nuclear ORF2 is present in the WT construct. However, for Fig5B this is difficult to judge by eye. In Fig. 2C the nuclear signal is displayed as nuclear fluorescent intensity. However, in the following figures this is delineated as nuclear/cyto intensitiy ratio, which makes a comparison difficult. Please harmonize the display of these results.

PLOS authors have the option to publish the peer review history of their article (what does this mean?). If published, this will include your full peer review and any attached files.

Reviewer #1: No

Reviewer #2: No
---

## [Decision Letter · Decision Letter 1]

10 Jun 2022

Dear Dr. Cocquerel,

Thank you very much for submitting your manuscript "An Arginine-Rich Motif in the ORF2 Capsid Protein Regulates the Hepatitis E Virus Lifecycle and Interactions with the Host Cell" for consideration at PLOS Pathogens. As with all papers reviewed by the journal, your manuscript was reviewed by members of the editorial board and by several independent reviewers. In light of the reviews (below this email), we would like to invite the resubmission of a revised version that takes into account the reviewers' comments. We would like you to focus on the major points raised by reviewer 1.

Your study is of considerable interest but these remaining points need to be addressed before we can make a final decision on your submission..

Sincerely,

Alexander Ploss, Ph.D.

Guest Editor

PLOS Pathogens

Guangxiang Luo

Section Editor

PLOS Pathogens

Kasturi Haldar

Editor-in-Chief

PLOS Pathogens

orcid.org/0000-0001-5065-158X

Michael Malim

Editor-in-Chief

PLOS Pathogens

orcid.org/0000-0002-7699-2064

Reviewer's Responses to Questions

**Part I - Summary**

Reviewer #1: The authors added new data in revised manuscript, but the following concerns are still not been addressed.

Reviewer #2: In their manuscript entitled “An Arginine-Rich Motif in the ORF2 Capsid Protein Regulates the Hepatitis E Virus Lifecycle and Interactions with the Host Cell” Hervouet et al. use an HEV cell culture model to elucidate the role of ARM on multiple steps of HEV propagation. Utilizing a series of ORF2 mutants expressed either in the p6 HEV strain or as ORF2 capsid protein they delineate a crucial role of the ARM as functional nuclear localization signal which seems to be important for viral production. With the help of specific inhibitors and co- localization studies, the authors identified a CRM1-dependent egress from the nucleus and 3 important nuclear export signals within the ORF2 protein. The main body of the manuscript focusses on the maturation and addressing of ORF2. The authors found evidence for a role of a furin-dependent secretion pathway in the maturation process. With the generation and evaluation of additional mutants including the signal and/or the encoding sequence of CD4 by state-of the art techniques the authors delineate a tight interplay of the ARM with the respective signal peptide for addressing, topology and egress of the capsid proteins. Bringing this data set together the authors finally delineate a model for the generation of the infectious or non-infectious forms of the HEV particles.

The manuscript is well written, the experiments are appropriate and on a technically high level. Following on the previous reports from the group this manuscript sheds light into the so far under investigated HEV life-cycle and the generation of the different capsid forms.

Overall, this manuscript describes several novelties in the field and sets the ground for several future scientific perspectives concerning HEV.

**Part II – Major Issues: Key Experiments Required for Acceptance**

Reviewer #1: 1) Subcellular localization of ORF2 does not change obviously during infection condition (Fig 1D) compared with electroporation (Fig 1A-C), suggesting that it may be an artifact in electroporation condition due to ORF2 overexpression.

2) Fig 2D, there are multiple rounds of infection already, so the phenotype could be due the alterations of ORF3. This concern is not been addressed. According to the result in S7 Fig, the direct conclusion is that deletion of ORF3 don’t impair ORF2 nuclear localization. Our question is how to ensure that the fewer infectious particles produced is indeed caused by ARM mutation rather than ORF3 mutation (ORF2 and ORF3 ORFs are largely overlapped)? To answer this question, I think the possible strategies are as follows: (1) When generating ARM mutation to impair ORF2 nuclear localization in the contents of genomic sequence, please make sure the ORF3 amino acids unchanged. (2) utilize the trans-complementation system to uncouple the expression of ORF2 and ORF3 protein, then investigate the ARM mutation on virus production.

3) From the total extract panel in new Fig 2, it is obvious that ORF2 stability is dramatically impaired of ΔSP1, but the authors didn’t mention it in response.

Firstly, in S7 Fig, a WB result is required to show that the deletion of ORF3 is successful. More importantly, this assay (S7 Fig) can’t answer our question.

Reviewer #2: This reviewer feels, that all questions have been adequately adressed by the authors in this revision.

**Part III – Minor Issues: Editorial and Data Presentation Modifications**

Reviewer #1: 1) Please check primers to construct 5R/5A mutation. It seems to be wrong.

Reviewer #2: This reviewer feels, that all questions have been adequately adressed by the authors in this revision.

PLOS authors have the option to publish the peer review history of their article (what does this mean?). If published, this will include your full peer review and any attached files.

Reviewer #1: No

Reviewer #2: No
---

## [Decision Letter · Decision Letter 2]

6 Aug 2022

Dear Pr Cocquerel,

We are pleased to inform you that your manuscript 'An Arginine-Rich Motif in the ORF2 Capsid Protein Regulates the Hepatitis E Virus Lifecycle and Interactions with the Host Cell' has been provisionally accepted for publication in PLOS Pathogens.

Best regards,

Alexander Ploss, Ph.D.

Guest Editor

PLOS Pathogens

Guangxiang Luo

Section Editor

PLOS Pathogens

Kasturi Haldar

Editor-in-Chief

PLOS Pathogens

orcid.org/0000-0001-5065-158X

Michael Malim

Editor-in-Chief

PLOS Pathogens

orcid.org/0000-0002-7699-2064

Reviewer Comments (if any, and for reference):

Reviewer's Responses to Questions

**Part I - Summary**

Reviewer #1: The authors performed additional experiments to address reviewers questions. Most of the concerns and questions have been addressed.

**Part II – Major Issues: Key Experiments Required for Acceptance**

Reviewer #1: (No Response)

**Part III – Minor Issues: Editorial and Data Presentation Modifications**

Reviewer #1: (No Response)

PLOS authors have the option to publish the peer review history of their article (what does this mean?). If published, this will include your full peer review and any attached files.

Reviewer #1: No

---

## [Editor Report · Acceptance letter]

22 Aug 2022

Dear Pr Cocquerel,

We are delighted to inform you that your manuscript, "An Arginine-Rich Motif in the ORF2 Capsid Protein Regulates the Hepatitis E Virus Lifecycle and Interactions with the Host Cell," has been formally accepted for publication in PLOS Pathogens.

Best regards,

Kasturi Haldar

Editor-in-Chief

PLOS Pathogens

orcid.org/0000-0001-5065-158X

Michael Malim

Editor-in-Chief

PLOS Pathogens

orcid.org/0000-0002-7699-2064